# Mitochondrial outer membrane integrity regulates a ubiquitin-dependent and NF-κB-mediated inflammatory response

Esmee Vringer [1,2,8], Rosalie Heilig [1,2,8], Joel S Riley [1,2,3], Annabel Black [1,2], Catherine Cloix[1,2], George Skalka[1,2], Alfredo E Montes-Gómez [1,2], Aurore Aguado [1,2], Sergio Lilla [2], Henning Walczak [4,5,6], Mads Gyrd-Hansen [7], Daniel J Murphy [1,2], Danny T Huang [1,2], Sara Zanivan[1,2] & Stephen WG Tait [1,2✉]

## Abstract

**Mitochondrial outer membrane permeabilisation (MOMP) is often essential for apoptosis, by enabling cytochrome *c* release that leads to caspase activation and rapid cell death. Recently, MOMP has been shown to be inherently pro-inflammatory with emerging cellular roles, including its ability to elicit anti-tumour immunity. Nonetheless, how MOMP triggers inflammation and how the cell regulates this remains poorly defined. We find that upon MOMP, many proteins localised either to inner or outer mitochondrial membranes are ubiquitylated in a promiscuous manner. This extensive ubiquitylation serves to recruit the essential adaptor molecule NEMO, leading to the activation of pro-inflammatory NF-κB signalling. We show that disruption of mitochondrial outer membrane integrity through different means leads to the engagement of a similar pro-inflammatory signalling platform. Therefore, mitochondrial integrity directly controls inflammation, such that permeabilised mitochondria initiate NF-κB signalling.**

**Keywords** Mitochondria; Cell Death; Inflammation; NF-κB; Ubiquitin
**Subject Categories** Immunology; Organelles; Signal Transduction

## Introduction

Apoptotic cell death is considered an immunosilent form of cell death, in line with it being a major type of homeostatic cell death. Mitochondrial outer membrane permeabilisation (MOMP) is often essential to initiate apoptosis by enabling cytochrome *c* release, leading to rapid caspase activation and cell death (Bock and Tait, 2020). Nonetheless, upon a lethal stress, MOMP commits a cell to die regardless of caspase activation through so-called caspase-independent cell death (CICD). This is due to widespread MOMP causing a catastrophic loss in mitochondrial function (Lartigue et al, 2009).

Recent research has revealed that MOMP is inherently pro-inflammatory (Giampazolias et al, 2017; Marchi et al, 2022). For instance, mitochondrial DNA (mtDNA) is released from permeabilised mitochondria through BAX/BAK macropores, leading to activation of cGAS-STING signalling and a type I interferon response (McArthur et al, 2018; Riley et al, 2018). Importantly, while wholly dispensable for cell death, caspase activity serves to inhibit inflammation during mitochondrial apoptosis. Caspases inhibit inflammation in dying cells through multiple means, including direct cleavage of pro-inflammatory signalling proteins such as cGAS, inhibition of protein translation and promoting rapid removal of dying cells via the exposure of "eat-me" signals (McIlwain et al, 2013; Ning et al, 2019; Ravichandran, 2011).

By enhancing MOMP-induced inflammation through caspase inhibition, we and others have shown that engaging CICD in tumour cells can lead to anti-tumour immunity dependent on cGAS-STING and NF-κB signalling in the dying cell (Giampazolias et al, 2017; Han et al, 2020). We also reported that MOMP can occur in a limited cohort of mitochondria—an event we termed minority MOMP—in the absence of cell death (Cao et al, 2022; Ichim et al, 2015). Minority MOMP can promote caspase-dependent DNA-damage. Intriguingly others have discovered that minority MOMP causes inflammation required for the restriction of bacteria. (Brokatzky et al, 2019). More recently, we have found that minority MOMP contributes to the inflammatory phenotype of senescent cells, thereby directly bridging apoptotic signalling with senescence (Victorelli et al, 2023).

Therefore, MOMP-induced inflammation—alongside having physiological functions—represents a therapeutic target in cancer. Nonetheless, how MOMP elicits inflammation and how this is restrained remains poorly defined. Previous studies have shown

[1]Cancer Research UK Scotland Institute, Switchback Road, Glasgow G61 1BD, UK. [2]School of Cancer Sciences, University of Glasgow, Switchback Road, Glasgow G61 1BD, UK. [3]Institute of Developmental Immunology, Biocenter, Medical University of Innsbruck, Innsbruck, Austria. [4]Centre for Cell Death, Cancer, and Inflammation (CCCI), UCL Cancer Institute, University College London, London, UK. [5]CECAD Cluster of Excellence, University of Cologne, Cologne, Germany. [6]Center for Biochemistry, Faculty of Medicine and University Hospital Cologne, University of Cologne, Cologne, Germany. [7]Department of Immunology and Microbiology, LEO Foundation Skin Immunology Research Center, University of Copenhagen, Copenhagen, Denmark. [8]These authors contributed equally: Esmee Vringer, Rosalie Heilig. ✉E-mail: stephen.tait@glasgow.ac.uk

that permeabilised mitochondria can be targeted to lysosomes dependent on canonical autophagy (Lindqvist et al, 2018). How permeabilised mitochondria are specifically targeted for degradation—potentially limiting inflammation following MOMP, is not known. We initially set out to address this question, finding that upon MOMP, mitochondria are extensively ubiquitylated. Mitochondrial ubiquitylation has been shown to serve as a signal for mitophagy, best evidenced in mitophagy promoted by the E3 ubiquitin ligase Parkin (Vargas et al, 2022). Surprisingly, we find that autophagy is not essential for mitochondrial degradation following MOMP. Upon further investigation, we found that MOMP-induced ubiquitylation of mitochondria serves as an inflammatory signal, recruiting the essential NF-κB signalling adaptor, NF-κB essential modulator (NEMO). In this way, mitochondrial outer membrane integrity dictates the initiation of an NF-κB inflammatory response, contributing to MOMP-induced inflammation.

## Results

### Permeabilised mitochondria are ubiquitylated and can be degraded independent of canonical autophagy

Following MOMP, autophagy targets permeabilised mitochondria for degradation and suppresses MOMP-induced inflammation (Colell et al, 2007; Lindqvist et al, 2018). Given this, our initial goal was to understand how MOMP triggers mitochondrial removal. To engage mitochondrial apoptosis, U2OS cells were treated with a combination of BH3-mimetics, ABT-737 (inhibits BCL-2, BCL-xL and BCL-w) and S63845 (inhibits MCL-1), then analysed for cell viability by SYTOX Green exclusion using Incucyte live-cell imaging. Combined BH3-mimetic treatment in wild-type U2OS cells led to rapid cell death that was inhibited by co-treatment with pan-caspase inhibitor Q-VD-OPh or CRISPR-Cas-9 mediated deletion of BAX and BAK, two proteins essential for MOMP, confirming engagement of mitochondrial apoptosis (Fig. EV1A,B). Using this approach, we next assessed mitochondrial content in U2OS cells following MOMP under conditions of CICD by using the combination treatment of ABT-737, S63845, and Q-VD-OPh. Mitochondrial content was determined by western blot for mitochondrial proteins or via qPCR for mitochondrial DNA (Fig. 1A,B). Reduction in cellular mitochondrial content was observed specifically following MOMP, as evidenced by a loss of mtDNA and mitochondrial protein content in a BAX/BAK-dependent manner (Fig. 1A,B). We next treated U2OS cells to undergo CICD and visualised mitochondria using MitoTracker Green (MTG) (Fig. 1C). At early time points post-treatment (3 h), mitochondria underwent fragmentation and peri-nuclear accumulation, whereas loss of MTG signal was observed at longer time points (24 h), consistent with mitochondrial degradation. Mitochondrial fragmentation, peri-nuclear accumulation and loss of MTG signal was absent in BAX/BAK deficient cells, demonstrating a requirement for MOMP (Fig. 1C). Mitochondrial ubiquitylation is a well-established signal for autophagic removal of mitochondria, a process called mitophagy (Vargas et al, 2022). Therefore, we investigated whether mitochondria are ubiquitylated upon MOMP. SVEC4-10 murine endothelial cells were treated to undergo CICD and mitochondrial-enriched fractions were probed for

ubiquitylation by western blot using a pan-ubiquitin antibody. Consistent with the engagement of MOMP, SMAC (also called DIABLO) was depleted from the mitochondrial-enriched fraction upon CICD. Importantly, while basal levels of mitochondrial ubiquitylation were detectable, an extensive increase of protein ubiquitylation was evident in the mitochondria-enriched fraction specifically following MOMP (Fig. 1D). The increase in mitochondrial ubiquitylation was dependent upon MOMP since protein ubiquitylation on the mitochondria remained consistent in BAX/BAK deficient cells following BH3-mimetic treatment (Fig. EV1C). To corroborate these findings, U2OS EMPTY^CRISPR and BAX/BAK^CRISPR cells were immunostained following induction of CICD using a combination of anti-ubiquitin and mitochondrial COXIV antibodies. Upon CICD, ubiquitin localised with mitochondria in U2OS EMPTY^CRISPR cells but not in U2OS BAX/BAK^CRISPR cells (Fig. 1E,F), consistent with the earlier mitochondrial fractionation experiment (Fig. EV1C). To investigate whether inhibition of caspase activity was required for the ubiquitylation of mitochondria following MOMP, SVEC4-10 cells were treated with BH3-mimetics with or without the pan-caspase inhibitor Q-VD-OPh. Western blot analysis of mitochondria-enriched fractions demonstrated increased ubiquitylation irrespective of caspase inhibition (Fig. EV1D).

Ubiquitylation can target organelles for autophagic degradation via the recruitment of specific autophagy adaptor molecules (Vargas et al, 2022). We therefore investigated whether autophagy was required for the degradation of mitochondria following MOMP by engaging CICD in U2OS cells deficient in ATG5 or ATG7, two proteins essential for canonical macroautophagy (Komatsu et al, 2005; Kuma et al, 2004). ATG5 and ATG7 loss, as well as functional autophagy deficiency, evident by an absence of lipidated LC3 (LC3 II), was confirmed via western blot (Fig. 1G). Surprisingly, treatment of cells with BH3-mimetics and caspase inhibitor caused a reduction of mitochondria (as determined by the loss of mitochondrial protein content) independent of autophagy (Fig. 1G). Finally, we investigated whether ubiquitylation and/or proteasomal function was required for the degradation of mitochondrial proteins following MOMP. U2OS were treated with BH3-mimetic/Q-VD-OPh for 7 or 24 h in the presence of TAK-243 (inhibiting the first step of ubiquitylation through E1 inhibition (Hyer et al, 2018)) or MG-132 (proteasome inhibitor) (Fig. EV1E–H). As expected, proteasome inhibition (MG-132 treatment) increased cellular ubiquitylation (Fig. EV1E), whereas E1 inhibition (TAK-243 treatment) effectively blocked ubiquitylation (Fig. EV1G). At 24 h following BH3-mimetic/Q-VD-OPh treatment, mitochondrial protein content was rescued upon proteasomal or E1 inhibition following MOMP (Fig. EV1F,H). These data demonstrate that following MOMP, mitochondria can be degraded independently of autophagy and at extended time points following MOMP, mitochondrial proteins are degraded in a ubiquitin–proteasome-dependent manner.

### Widespread mitochondrial protein ubiquitylation occurs upon MOMP

We next characterised mitochondrial protein ubiquitylation upon MOMP. Di-glycine remnant proteomics can identify ubiquitylated proteins by immunoprecipitation of di-Gly-motifs left on ubiquitylated proteins after trypsinisation (Xu et al, 2010). Using this

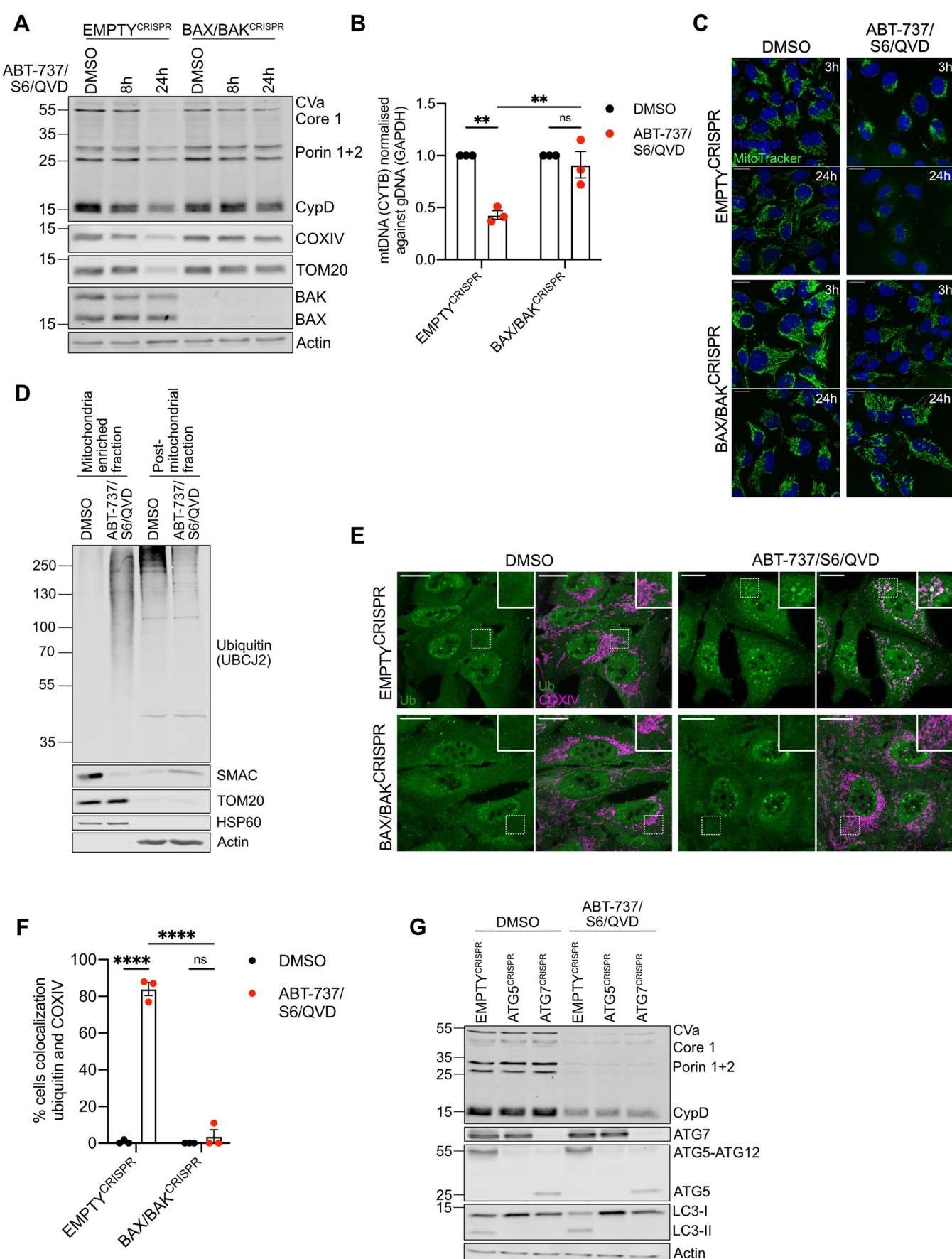

**Figure 1. Mitochondrial depletion after MOMP does not require autophagy.**

(A) U2OS EMPTY[CRISPR] and BAX/BAK[CRISPR] cells treated with 10 μM ABT-737, 2 μM S63845 and 20 μM Q-VD-OPh for 8 or 24 h. Mitochondrial depletion was assessed by blotting for several mitochondrial proteins. (B) U2OS EMPTY[CRISPR] and BAX/BAK[CRISPR] cells were treated with 10 μM ABT-737, 2 μM S63845 and 20 μM Q-VD-OPh for 24 h. Graphs shows presence of mtDNA relative to genomic DNA in $n = 3$ independent experiments, error bars represent s.e.m. (C) U2OS EMPTY[CRISPR] and BAX/BAK[CRISPR] cells were treated with 10 μM ABT-737, 2 μM S63845 and 20 μM Q-VD-OPh for 3 or 24 h. Nuclei were stained using Hoechst, and mitochondria with Mitotracker Green. Images are representative of three independent experiments. Scale bar is 20 μm. (D) SVEC4-10 cells treated for 1 h with 10 μM ABT-737, 10 μM S63845 and 30 μM Q-VD-OPh. Mitochondria were isolated using Dounce homogeniser. Lysates for blotted for ubiquitin (UBCJ2), SMAC, TOM20, HSP60 and Actin. (E) U2OS EMPTY[CRISPR] and BAX/BAK[CRISPR] cells treated for 3 h with 10 μM ABT-737, 2 μM S63845 and 20 μM Q-VD-OPh. Cells were stained for ubiquitin (FK2) and mitochondrial COXIV. Images are representative of three independent experiments. Images are maximum projections of Z-stacks with a scale bar of 20 μm. (F) Quantification of (E) showing the percentage of cells with mitochondrial localised ubiquitin puncta. Graph displays mean values ± s.e.m. (error bar) of $n = 3$ independent experiments. (G) U2OS EMPTY[CRISPR], ATG5[CRISPR] and ATG7[CRISPR] expressing YFP-Parkin were treated with 10 μM ABT-737, 2 μM S63845 and 20 μM Q-VD-OPh for 24 h. Mitochondrial depletion was assessed by blotting for various mitochondrial proteins. Data information: (A,D,G) blots are representative of three independent experiments. Statistics for all experiments were performed using two-way ANOVA with Tukey correction. *$P < 0.05$, **$P < 0.01$, ****$P < 0.0001$. Source data are available online for this figure.

method, we investigated the ubiquitylome of SVEC4-10 cells treated to undergo CICD. Mass spectrometry proteomic analysis revealed a significant change in the ubiquitylome of CICD-treated SVEC4-10 cells compared to untreated (Fig. 2A). Gene-ontology (GO) term analysis and manual curation of proteins using MitoCarta 3.0 (Rath et al, 2021) revealed that most peptides (approx. 80%) that gained a ubiquitin modification after MOMP were mitochondrially localised (Fig. 2B–D; Appendix Table S1). Ubiquitylated mitochondrial proteins were not confined to one mitochondrial compartment, with broadly similar numbers of ubiquitylated proteins characterised as being localised to the mitochondrial outer membrane or mitochondrial inner membrane (Fig. 2B; Appendix Table S1). Notably, some proteins with increased ubiquitylation have been defined as being localised to the mitochondrial matrix, possibly reflecting mitochondrial inner membrane permeabilisation that we and others have reported previously (Fig. 2B; Appendix Table S1) (McArthur et al, 2018; Riley et al, 2018). These data demonstrate promiscuous ubiquitylation of mitochondrial proteins following MOMP.

## Mitochondrial protein ubiquitylation encompasses K63- and M1-ubiquitin linkages

Protein ubiquitylation is highly complex with specific ubiquitin linkages conferring distinct biological functions. For instance, K48-ubiquitin linkages are typically associated with targeting proteins for proteasomal degradation, whereas K63-ubiquitylation has signalling functions (Komander and Rape, 2012). Given this, we investigated the type of ubiquitin linkages that MOMP triggers. SVEC4-10 cells were treated to undergo CICD, and the mitochondrial-enriched fraction was blotted for K48- and K63-ubiquitin linkages using linkage-specific antibodies (Fig. 3A). This revealed an increase in K63-linked ubiquitin, but not K48-linked ubiquitin, in the mitochondrial fraction specifically during CICD. K63-linked ubiquitylation of mitochondria was also detected upon CICD by immunofluorescence (Fig. 3B,C). Finally, we made use of GFP-fused ubiquitin-binding domains (UBDs) developed to specifically visualise K63- and linear M1-ubiquitin linkages (Hrdinka et al, 2016). Consistent with our previous data, extensive K63-linked ubiquitin was detected on mitochondria following CICD (Fig. 3D,E). In contrast, mitochondrial localisation of M1-specific UBDs was observed in a smaller percentage of cells analysed. These data reveal that upon MOMP, mitochondrial proteins are enriched in K63- and M1-linked ubiquitin.

## Mitochondrial ubiquitylation recruits the essential NF-κB adaptor NEMO promoting NF-κB activation

We next sought to understand the potential biological functions of mitochondrial ubiquitylation following MOMP. Our previous data demonstrated that NF-κB is activated following MOMP, contributing to anti-tumorigenic effects of CICD (Giampazolias et al, 2017). This finding, coupled to the well-established connection between ubiquitylation and inflammatory signalling (Madiraju et al, 2022), led us to investigate if mitochondrial ubiquitylation may be involved in NF-κB activation during CICD. NEMO, an essential adaptor protein in canonical NF-κB signalling, initiates NF-κB activity through ubiquitin binding. Given this, we examined the localisation of NF-κB essential modulator (NEMO) under conditions of MOMP, by expressing GFP-NEMO in U2OS cells and treating with BH3-mimetic plus caspase inhibitor (Fig. 4A,B). Live-cell microscopy revealed rapid recruitment of NEMO to mitochondria following MOMP (Fig. 4A,B; Movie EV1—DMSO and Movie EV2—ABT-737/S6/QVD). Immunoblot analysis also demonstrated accumulation of endogenous NEMO on mitochondrial-enriched fractions specifically following MOMP (Figs. 4C and EV2A). Importantly, mitochondrial translocation of GFP-NEMO occurred upon MOMP in a BAX/BAK-dependent manner (Fig. 4D,E). To investigate if ubiquitylation was required for mitochondrial recruitment of NEMO we used the E1 inhibitor TAK-243 to block ubiquitylation. TAK-243 treatment effectively blocked mitochondrial ubiquitylation and mitochondrial recruitment of GFP-NEMO, (Fig. EV2B,C). In contrast, blocking the ubiquitin-like modification neddylation, using NAE1 inhibitor MLN4924 (Soucy et al, 2009), did not result in reduced ubiquitylation and GFP-NEMO translocation in SVEC4-10 cells (Fig. EV2D–F).

M1-ubiquitin linkages are often essential for NEMO activation, by binding to the UBAN domain on NEMO (Rahighi et al, 2009). In addition, the C-terminal zinc finger (ZF) domain of NEMO can enhance the binding of K63-ubiquitin chains to the UBAN (Cordier et al, 2009; Laplantine et al, 2009). To determine which domain(s) are required for ubiquitin-dependent recruitment after MOMP mutant versions of NEMO were created. NEMO mutants disrupting the ability of the UBAN (D311N) or C-terminus (ΔZF) to bind ubiquitin were expressed in U2OS cells. Both non-ubiquitin-binding mutants of NEMO failed to be recruited to the mitochondria after MOMP (Fig. 4F,G), suggesting that binding to K63-ubiquitin chains is required. To extend these findings, we made use of murine embryonic fibroblasts (MEFs) deficient in

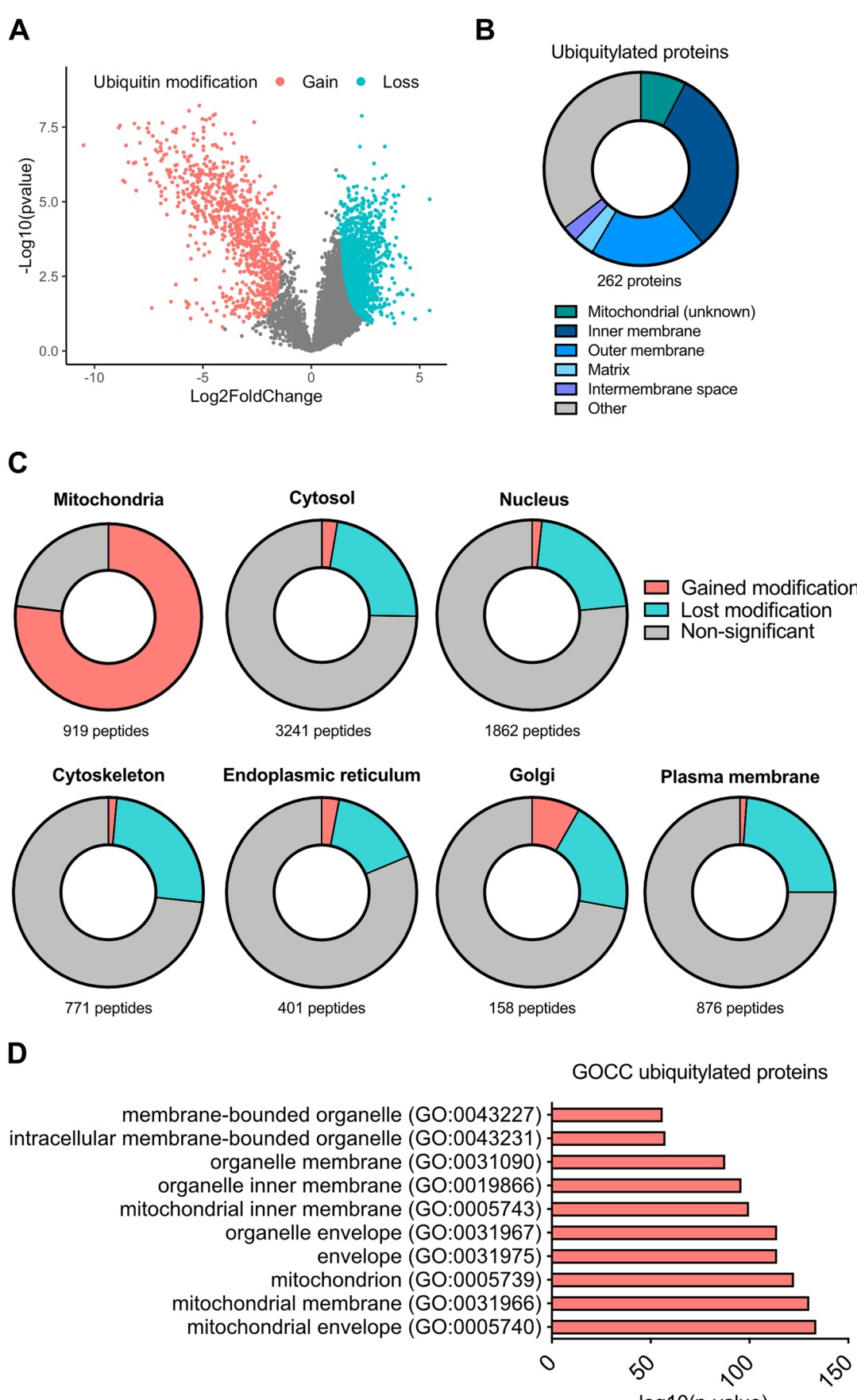

**A**

Ubiquitin modification ● Gain ● Loss

**B**

Ubiquitylated proteins

262 proteins

- Mitochondrial (unknown)
- Inner membrane
- Outer membrane
- Matrix
- Intermembrane space
- Other

**C**

Mitochondria — 919 peptides

Cytosol — 3241 peptides

Nucleus — 1862 peptides

- Gained modification
- Lost modification
- Non-significant

Cytoskeleton — 771 peptides

Endoplasmic reticulum — 401 peptides

Golgi — 158 peptides

Plasma membrane — 876 peptides

**D**

GOCC ubiquitylated proteins

- membrane-bounded organelle (GO:0043227)
- intracellular membrane-bounded organelle (GO:0043231)
- organelle membrane (GO:0031090)
- organelle inner membrane (GO:0019866)
- mitochondrial inner membrane (GO:0005743)
- organelle envelope (GO:0031967)
- envelope (GO:0031975)
- mitochondrion (GO:0005739)
- mitochondrial membrane (GO:0031966)
- mitochondrial envelope (GO:0005740)

-log10(p-value)

◄  **Figure 2.  Widespread ubiquitylation of mitochondrial proteins after MOMP.**

(A) Volcano plot of ubiquitylated proteins in SVEC4-10 cells treated for 3 h with 10 µM ABT-737, 10 µM S63845 and 30 µM Q-VD-OPh. The experiment performed with $n = 4$ independent repeats. Statistical analysis determining significance (coloured dots) was through using Student's $t$ test. Plot generated in RStudio. (B) Pie chart of ubiquitylated peptides categorised into mitochondrial compartments. Categorisation of peptides was performed using MitoCarta 3.0, UniProt and ProteinAtlas. (C) Cellular distribution of all hits from the isolated ubiquitin remnant-containing peptides. Categorisation using MitoCarta 3.0, UniProt and ProteinAtlas. (D) GO-term cellular compartment analysis of proteins with increased ubiquitylation after MOMP. Statistical analysis was performed with Fisher's exact test with corrected with false discovery rate. Graphs shows the top ten most significant hits. Source data are available online for this figure.

HOIP, the catalytic subunit of LUBAC E3 ligase complex required for M1-linked ubiquitylation (Peltzer et al, 2014). Importantly, mitochondrial recruitment of GFP-NEMO was not impaired in HOIP-deficient MEFs (Fig. EV3A,B). These data demonstrate that K63-linked ubiquitylation, but not LUBAC-dependent M1-linked ubiquitylation, is required for NEMO recruitment to the mitochondria.

We next investigated if mitochondrial recruitment of NEMO facilitates NF-κB activation. SVEC4-10 cells expressing wild-type and non-ubiquitin-binding variants of GFP-NEMO (D311N and ΔZF) were treated to engage CICD and NF-κB activation was determined by nuclear NF-κB p65 translocation. In contrast to wild-type NEMO, both non-ubiquitin-binding variants of NEMO significantly inhibited NF-κB activation as determined by a reduction in nuclear p65 (Fig. 4H,I). Similar experiments were performed in SVEC4-10 cells treated with siRNA to deplete endogenous murine NEMO (Fig. EV3C–E). Depletion of NEMO in parental SVEC4-10 cells completely abolished nuclear p65 translocation. As expected, this was rescued by ectopic expression of human GFP-NEMO. In contrast, expression of human GFP-NEMOD311N failed to restore NF-κB p65 nuclear translocation, agreeing with our previous data. These data support a model whereby K63-ubiquitylation of mitochondria following MOMP enables NEMO recruitment leading to NF-κB activation.

## Ubiquitin-dependent NF-κB activation after MOMP is independent of canonical mitochondrial E3 ligases

Through MOMP, our findings directly link mitochondrial integrity to mitochondrial ubiquitylation and pro-inflammatory signalling. We next sought to identify which ubiquitin E3 ligase(s) may be responsible for mitochondrial ubiquitylation. One candidate is the E3 ligase Parkin, since active Parkin causes widespread ubiquitylation of mitochondrial proteins (Sarraf et al, 2013). Nonetheless, both SVEC4-10 and U2OS cells used in our studies do not express detectable Parkin (Fig. 5A), arguing that mitochondrial ubiquitylation following MOMP does not require Parkin. Parkin activity requires the kinase PINK1. PINK1 can also activate alternative E3 ligases such as ARIH1 (Villa et al, 2017). To investigate a potential role for PINK1, we generated PINK1CRISPR SVEC4-10 cell lines (Fig. EV4A). Confirming functional deletion, cells lacking PINK1 failed to recruit YFP-Parkin to mitochondria following CCCP treatment, in contrast to EMPTYCRISPR cells (Fig. EV4B). YFP-Parkin was not recruited to mitochondria after MOMP irrespective of PINK1 deletion (Fig. EV4B). Importantly, recruitment of GFP-NEMO was not impaired by the deletion of PINK1 (Fig. 5B,C), indicating that PINK1 does not have a role in ubiquitin-dependent recruitment of NEMO after MOMP.

The mitochondrial resident E3 ligases MUL1 (also called MAPL) and MARCH5 have roles in various cellular processes

such as mitochondrial dynamics, protein import, cell death and inflammation (Braschi et al, 2009; Haschka et al, 2020; Phu et al, 2020; Shiiba et al, 2020). Notably, increased ubiquitylation of both MUL1 and MARCH5 was detected upon MOMP (Appendix Table S1). Single and double knockout MUL1CRISPR and MARCH5CRISPR SVEC4-10 cell lines were generated (Fig. EV4C,D), however no differences in IκBα phosphorylation or IκBα degradation, two indicators of NF-κB activation, were observed (Fig. 5D). Moreover, no impact on mitochondrial ubiquitylation and GFP-NEMO recruitment following MOMP was observed in SVEC4-10 MUL1,MARCH5CRISPR cells (Fig. 5E,F). Interestingly, MARCH5 is degraded upon MOMP (Fig. 5D) indicating that its ubiquitylation observed in the ubiquitin remnant proteomics study might be linked to proteasomal degradation (Appendix Table S1).

The E3 ligase XIAP was previously described for its involvement in the recruitment of endolysosomes through ubiquitylation of mitochondrial proteins after MOMP (Hamacher-Brady et al, 2014). XIAPCRISPR SVEC4-10 cell lines were generated to validate the importance of XIAP in mitochondrial-driven inflammation (Fig. EV4E). No differences were observed in expression of pro-inflammatory cytokines after MOMP (Fig. EV4F), despite observing a small reduction in the percentage of cells with mitochondrial ubiquitylation and GFP-NEMO recruitment (Fig. 5G,H).

We have previously found that MOMP can elicit NIK-dependent NF-κB activity, therefore we investigated whether mitochondrial recruitment of NEMO was NIK-dependent. NIK-deleted SVEC4-10 cells expressing GFP-NEMO were generated by CRISPR/Cas-9 genome editing (Fig. EV4G) and treated to undergo CICD. NEMO translocated to mitochondria independent of NIK expression (Fig. 5I,J).

Combined, these data demonstrate that established mitochondrial E3 ligases and NIK activation are not required for mitochondrial ubiquitylation or NEMO recruitment following MOMP.

## Ubiquitin-dependent mitochondrial inflammation is regulated by mitochondrial outer membrane integrity

We aimed to understand how MOMP might trigger mitochondrial ubiquitylation leading to NEMO recruitment. MOMP has been reported to trigger multiple effects including loss of mitochondrial respiratory chain function, induction of reactive oxygen species (ROS), increased calcium uptake and mitochondrial permeability transition (MPT). We therefore investigated if these processes were sufficient to trigger mitochondrial ubiquitylation and NEMO recruitment. SVEC4-10 cells expressing GFP-NEMO were treated with rotenone, antimycin A or oligomycin D-complex I, III and ATP synthase inhibitors respectively. Alternatively, cells were treated to erastin to induce mitochondrial calcium uptake or treated to undergo CICD in the presence of cyclosporin A to inhibit MPT. Cells were analysed for mitochondrial ubiquitylation and

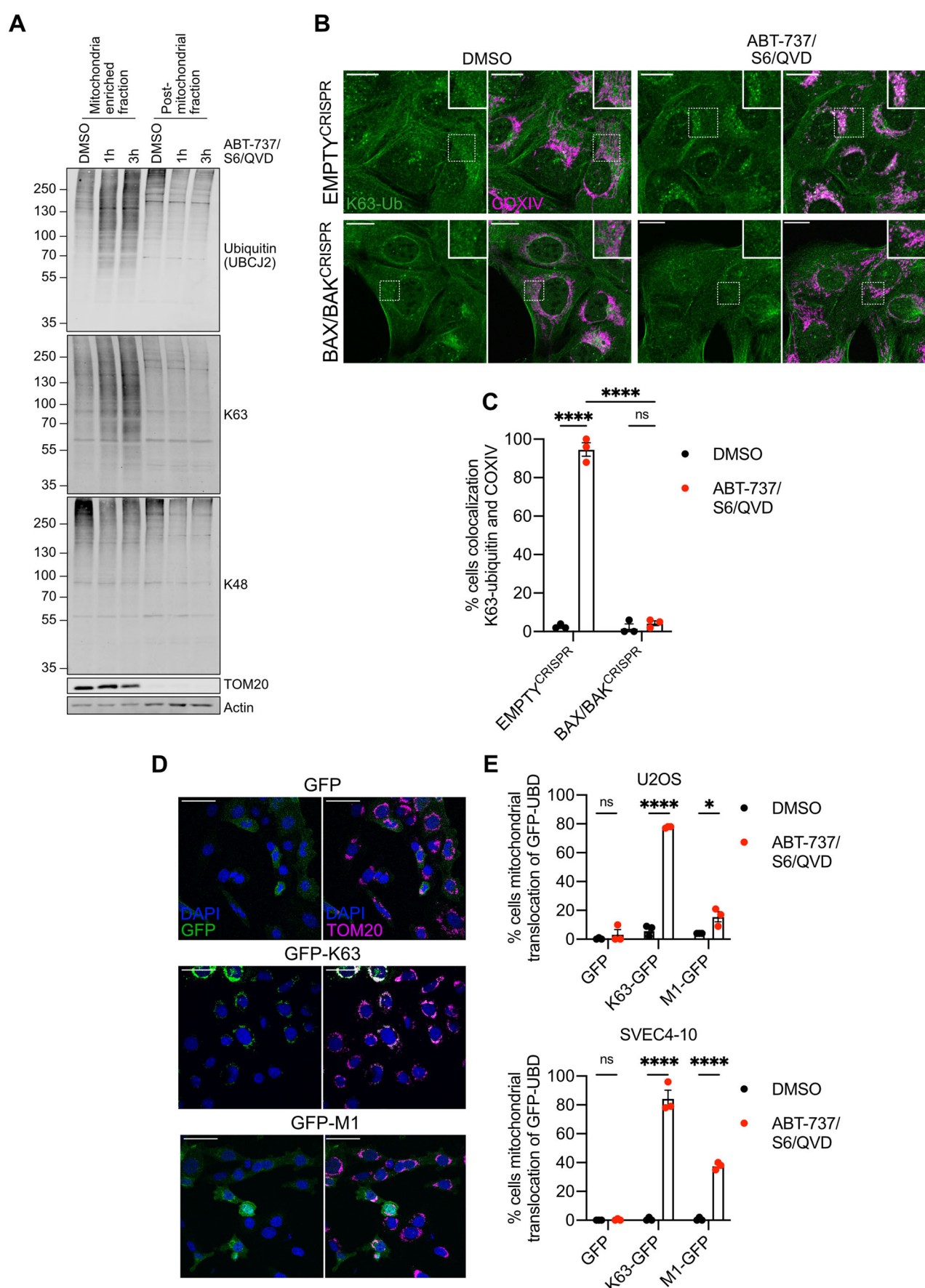

◀ **Figure 3.  K63-linked ubiquitylation on mitochondria after MOMP.**

(A) SVEC4-10 cells treated with for 1 or 3 h with 10 μM ABT-737, 10 μM S63845 and 30 μM Q-VD-OPh. Mitochondria were isolated using digitonin fractionation buffer and antibodies against ubiquitin (UBCJ2), K63- and K48-specific ubiquitin were used. Blots representative for three independent experiments. (B) U2OS EMPTY[CRISPR] and BAX/BAK[CRISPR] cells treated with 10 μM ABT-737, 2 μM S63845 and 20 μM Q-VD-OPh for 3 h. Stained for K63-ubiquitin and mitochondrial COXIV. Images are maximum projections of Z-stacks with a scale bar of 20 μm and are representative of three independent experiments. (C) Quantification of (B) showing the percentage of cells with mitochondrial localised K63-ubiquitin puncta. Statistics performed using two-way ANOVA with Tukey correction. (D) SVEC4-10 cells expressing doxycycline-inducible K63 or M1-UBDs. Cells were treated for 1 h with 10 μM ABT-737, 10 μM S63845 and 30 μM Q-VD-OPh. Images are representative of three independent experiments with a scale bar of 50 μm. (E) Quantification of (D) showing the percentage of SVEC4-10 cells with mitochondrial localised GFP-UBDs. Also includes the quantification of U2OS cells expressing doxycycline-inducible K63- or M1-UBDs treated for 3 h with 10 μM ABT-737, 2 μM S63845 and 20 μM Q-VD-OPh. Statistics were performed using multiple unpaired $t$ tests. Data information: (C, E) graphs display mean values ± s.e.m. (error bars) of $n = 3$ independent experiments. **$P < 0.01$, ***$P < 0.001$. ****$P < 0.0001$. Source data are available online for this figure.

NEMO translocation (Fig. EV5A,B,E,F). Rotenone led to an expected increase in mitochondrial ROS and erastin to increased mitochondrial calcium uptake (Fig. EV5C,D). In all cases, pharmacological manipulation of these described processes failed to promote mitochondrial ubiquitylation or NEMO recruitment.

Given this, we next sought to define if pro-inflammatory mitochondrial ubiquitylation was specific to mitochondrial apoptosis or initiated due to loss of mitochondrial integrity. For this purpose, we used the compound raptinal that can cause MOMP independent of BAX and BAK (Heimer et al, 2019; Palchaudhuri et al, 2015). In agreement, BAX/BAK deficient SVEC4-10 cells were protected against cell death induced by BH3-mimetics but remained sensitive to raptinal-induced cell death in a caspase-dependent manner (Fig. EV5G,H). We next investigated GFP-NEMO translocation and mitochondrial ubiquitylation following raptinal treatment in BAX/BAK-deleted SVEC4-10 cells. Importantly, raptinal treatment led to robust mitochondrial ubiquitylation and GFP-NEMO translocation independently of BAX and BAK (Fig. 6A,B). Consistent with this, nuclear translocation of p65 was also observed in BAX/BAK-deleted cells following raptinal treatment (Fig. 6C,D). Finally, increased transcription of NF-κB targets *Kc* and *Tnf* was detected following raptinal treatment in BAX/BAK-deleted SVEC4-10 cells (Figs. 6E and EV5I). Congruent with earlier findings, BH3-mimetic-induced ubiquitylation, NEMO translocation and NF-κB activity required BAX and BAK (Fig. 6A–E). These data demonstrate that loss of mitochondrial outer membrane integrity is sufficient to induce mitochondrial ubiquitylation leading to NEMO recruitment and an NF-κB-dependent inflammatory response.

## Discussion

We find that mitochondria are promiscuously ubiquitylated upon disruption of mitochondrial outer membrane integrity. Numerous proteins localising to both outer and inner mitochondrial membranes were found to be ubiquitylated. By investigating the functions of mitochondrial ubiquitylation after MOMP we unexpectedly found that degradation of mitochondria could occur independently of canonical autophagy. We found that mitochondrial ubiquitylation directly promotes inflammatory NF-κB activation through mitochondrial recruitment of the adaptor molecule NEMO. These data connect mitochondrial outer membrane integrity to direct activation of NF-κB activity, contributing to the pro-inflammatory effects of MOMP.

Given the bacterial ancestry of mitochondria, our findings raise striking parallels with cell-intrinsic responses to bacterial infection. For instance, ubiquitylation of intracellular *Salmonella* Typhimurium

serves as a platform to initiate pro-inflammatory NF-κB signalling as an innate immune response (Noad et al, 2017; van Wijk et al, 2017). Notably, the mitochondrial inner membrane and bacterial membranes share similarities, for instance enrichment in cardiolipin (Vringer and Tait, 2022). We speculate that upon cytosolic exposure, the mitochondrial inner membrane may represent a damage-associated molecular pattern (DAMP) eliciting ubiquitylation, NEMO recruitment and inflammation. Nonetheless, distinct differences exist between NEMO recruitment leading to NF-κB activation on invading bacteria and permeabilised mitochondria. The most striking distinction is that, unlike bacteria, LUBAC-dependent M1-linked ubiquitylation is not required for the recruitment of NEMO to permeabilised mitochondria. This is best evidenced by mitochondrial recruitment of NEMO in cells deficient in HOIP, the catalytic subunit of LUBAC complex required for M1-linked ubiquitylation. Instead, NEMO recruitment to mitochondria appears dependent on its ability to bind K63-ubiquitylated proteins, as we observe extensive K63-linked (but not degradative K48-linked) mitochondrial ubiquitylation upon MOMP. Interestingly, a recent study has shown that mitochondria amplify TNF-induced NF-κB signalling (Wu et al, 2022). In this paradigm, the mitochondrial outer membrane serves as a platform for LUBAC activity enhancing linear M1-linked ubiquitylation of NEMO. Together with our data, this positions mitochondria in different contexts as both initiators and amplifiers of NF-κB-dependent signalling.

Mechanistic questions remain—not least the identity of the ubiquitin E3 ligase(s) required for MOMP-induced ubiquitylation. Our data argue against any role for PINK1/Parkin, XIAP or resident mitochondrial ubiquitin ligases such as MARCH5 and MUL1. Secondly, what properties of permeabilised mitochondria that initiate ubiquitylation remain unknown. Targeted engagement of processes associated with MOMP including loss of mitochondrial respiratory function, ROS induction and calcium uptake failed to cause mitochondrial ubiquitylation and NEMO recruitment. Importantly, our data shows that mitochondrial ubiquitylation occurs upon loss of mitochondrial outer membrane integrity, independent of how this is achieved. This is best evidenced by MOMP engaged by either BAX/BAK or using the drug raptinal (in BAX/BAK null cells) as both cause mitochondrial ubiquitylation, NEMO recruitment and NF-κB activation. Although speculative at this point, possibly proteins located on the inner mitochondrial membrane, when exposed to the cytosol recruit and activate cytosolic ubiquitin ligases.

Our initial premise for this study stemmed from the hypothesis that mitochondrial ubiquitylation may serve as targeting signal for mitophagy, akin to PINK1/Parkin-mediated mitophagy. The kinetics of mitochondrial degradation were significantly slower than NEMO recruitment to mitochondria and NF-κB activation, thus mitochondrial degradation is unlikely to impact this inflammatory pathway. At late time points

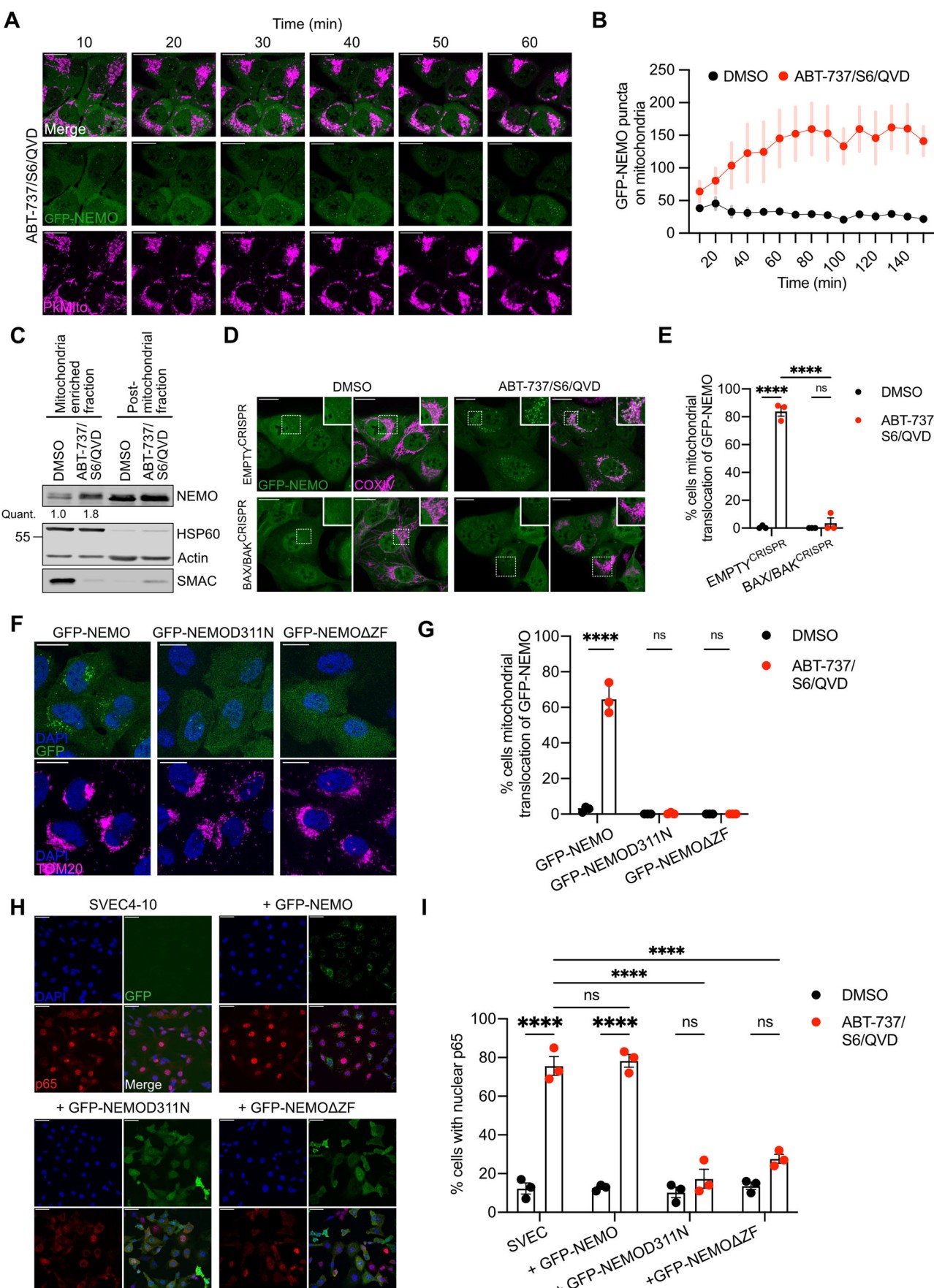

◄

**Figure 4. Ubiquitin-dependent recruitment of NEMO to mitochondria is essential for NF-κB activation after MOMP.**

(A) Timelapse of U2OS cells expressing GFP-NEMO. Cells were treated with 10 μM ABT-737, 10 μM S63845 and 30 μM Q-VD-OPh. Mitochondria and nuclei are visualised using PkMito DeepRed and Hoechst, respectively. Cells were treated for 1 h with images taken every 10 min. Image is representative for three independent experiments. Scale bar is 20 μm. (B) U2OS cells expressing GFP-NEMO treated with treated with 10 μM ABT-737, 10 μM S63845 and 30 μM Q-VD-OPh were quantified for mitochondrial localised GFP-NEMO puncta over time. Puncta are calculated using ImageJ/Fiji "trainable Weka segmentation plug-in". The graph is representative of three biological repeats and shows the mean $+/-$ s.e.m. (error bars) of five fields of view taken over time. (C) SVEC4-10 cells treated for 3 h with 10 μM ABT-737, 10 μM S63845 and 30 μM Q-VD-OPh. Mitochondria were isolated using Dounce homogeniser and cellular fractions were probed with relevant antibodies. Mitochondrial localised NEMO was quantified normalising to mitochondrial content defined by HSP60 signal. (D) U2OS EMPTY$^{CRISPR}$ and BAX/BAK$^{CRISPR}$ cells expressing GFP-NEMO were treated for 3 h with 10 μM ABT-737, 2 μM S63845 and 20 μM Q-VD-OPh. Cells were immunostained for mitochondrial COXIV. Scale bar is 20 μm. Images are maximum projections of Z-stacks and are representative for three independent experiments. (E) Quantification of (D) showing the percentage of cells with mitochondrial localised GFP-NEMO puncta. (F) U2OS cells expressing GFP-NEMO, GFP-NEMOD311N or GFP-NEMOΔZF were treated for 3 h with 10 μM ABT-737, 2 μM S63845 and 20 μM Q-VD-OPh. Cells were immunostained for mitochondrial TOM20 and DAPI. Scale bar is 20 μm and images are representative for three independent experiments. (G) Quantification of (F) showing the percentage of cells with mitochondrial translocation of GFP-NEMO. (H) Parental SVEC4-10 cells and SVEC4-10 cells expressing GFP-NEMO, GFP-NEMOD311N or GFP-NEMOΔZF were treated for 1 h with 10 μM ABT-737, 10 μM S63845 and 30 μM Q-VD-OPh. Cells were immunostained for p65 and DAPI. Scale bar is 50 μm and images are representative for three independent experiments. (I) Quantification of (H) showing the GFP+ cells with nuclear translocation of p65. Data information: (E, G, I) graphs display mean values ± s.e.m. (error bars) of $n = 3$ independent experiments. Statistics are performed using two-way ANOVA with Tukey correction. **$P < 0.01$. ***$P < 0.001$. ****$P < 0.0001$. Source data are available online for this figure.

following MOMP, we found that mitochondrial proteins could be degraded in a ubiquitin–proteasome-dependent manner. This argues against a significant role for mitochondrial degradation in the suppression of MOMP-dependent NF-κB activity. Surprisingly, we found that mitochondrial degradation occurred in cells deficient in canonical autophagy. While this doesn't negate a role for autophagy in promoting removal of permeabilised mitochondria, it demonstrates that autophagy is not essential. Notably, we and others have previously found that upon MOMP, the mitochondrial outer membrane can be completely lost leaving what we called mito-corpses (Ader et al, 2019; Riley et al, 2018). Whether autophagy-independent degradation of permeabilised mitochondria occurs in a regulated manner remains an open question.

In summary, our data reveal a novel direct connection between mitochondrial function and engagement of inflammation, where disruption of mitochondrial integrity initiates pro-inflammatory NF-κB signalling through extensive ubiquitylation and NEMO recruitment. Intriguingly, the E3 ligase Parkin—activated by loss of mitochondrial respiratory function—has recently been found to promote NF-κB activation through mitochondrial ubiquitylation enabling NEMO recruitment (Harding et al, 2023). Therefore, both mitochondrial integrity and function can regulate NF-κB-dependent inflammation through multiple pathways. Given the numerous emerging functions of MOMP-induced inflammation, ranging from senescence to innate and anti-tumour immunity, basic understanding of this process may reveal new therapeutic opportunities.

## Methods

### Cell culture and chemicals

HEK293FT, SVEC4-10, MEFs and U2OS cells were cultured in high glucose DMEM supplemented with 10% FBS (Gibco #10438026), 2 mM glutamine (Gibco #25030081) and 1 mM sodium pyruvate (Gibco #11360070). Cells were cultured in 21% $O_2$ and 5% $CO_2$ at 37 °C. MEF $Tnf^{-/-}$ $Hoip^{+/+}$ and MEF $Tnf^{-/-}$ $Hoip^{-/-}$ cell lines have been described before (Peltzer et al, 2014). SVEC4-10 cells were purchased from ATCC. All cell lines were routinely tested for mycoplasma.

The following chemicals were used in this study: ABT-737 (APEXBIO #A8193), S63845 (Chemgood #C-1370), Q-VD-OPh (AdooQ Bioscience #A14915-25), Doxycycline hyclate (Sigma-Aldrich #D9891), erastin (Biotechne #5449/10), TAK-243

(MedChemExpress #HY-100487), MLN4924 (Selleck Chemical #S7109), MG-132 (Selleck Chemical #S2619), and raptinal (Millipore Sigma #SML1745), MitoTracker Green FM (Invitrogen #7514), PKmito DeepRed (SPIROCHROME #SC055), oligomycin (Sigma-Aldrich # O4876), antimycin A (Sigma-Aldrich #A8674), rotenone (Sigma-Aldrich #R8875), cyclosporin A (Sigma-Aldrich # 30024).

### Viral transduction

Overexpression and CRISPR cell lines were generated using lenti- or retroviral infection. For lentiviral transfections, 1 μg VSVG (Addgene #8454) and 1.86 μg psPAX2 (Addgene #12260) were used. For retroviral transfections, 1 μg VSVG and 1.86 μg HIV gag-pol (Addgene #14887) were used. For both transfections, 5 μg of plasmid was used. HEK293FTs were transfected using lipofectamine 2000 or lipofectamine 3000 according to the manufacturer's instructions. After 2 days, virus-containing media was removed from the HEK293FTs and supplemented with 10 μg/ml polybrene before transferring to target cells. Two days later, infection cells were selected using 2 μg/mL puromycin, 10 μg/mL blasticidin or 800 μg/mL neomycin. Some U2OS and SVEC4-10 lines expressing GFP were sorted for GFP expression instead of antibiotic selection.

The M6PblastGFP-NEMO, PMD-OGP and PMD-VSVG plasmids were gifted by Felix Randow, LMB Cambridge. The pLenti-CMV-TetRepressor, pDestination-eGFP-NES, pDestination-eGFP-SK63-NES and pDestination-eGFP-NCM1-NES plasmids were published previously (Hrdinka et al, 2016). CRISPR cell lines were generated using lentiCRISPRv1 or lentiCRISPRv2 vector (Addgene #52961) containing puromycin, blasticidin and neomycin resistance.

Human *ATG5* 5'-AAGAGTAAGTTATTTGACGT-3'
Human *ATG7* 5'-GAAGCTGAACGAGTATCGGC-3'
Human *BAK* 5'-GCCATGCTGGTAGACGTGTA-3'
Human *BAX* 5'-AGTAGAAAAGGGCGACAACC-3'
Mouse *Bak* 5'-GCGCTACGACACAGAGTTCC-3'
Mouse *Bax* 5'-CAACTTCAACTGGGGCCGCG-3'
Mouse *March5* 5'-AAGTACTCGGCGTTGCACTG-3'
Mouse *Mul1* 5'-TATATGGAGTACAGTACGG-3'
Mouse *Nik* 5'-TCGCTGGCCAGCGATCGCTC-3'
Mouse *Pink1* 5'-CTGATCGAGGAGAAGCAGG-3'
Mouse *Xiap* 5'-CATCAACATTGGCGCGAGCT-3'

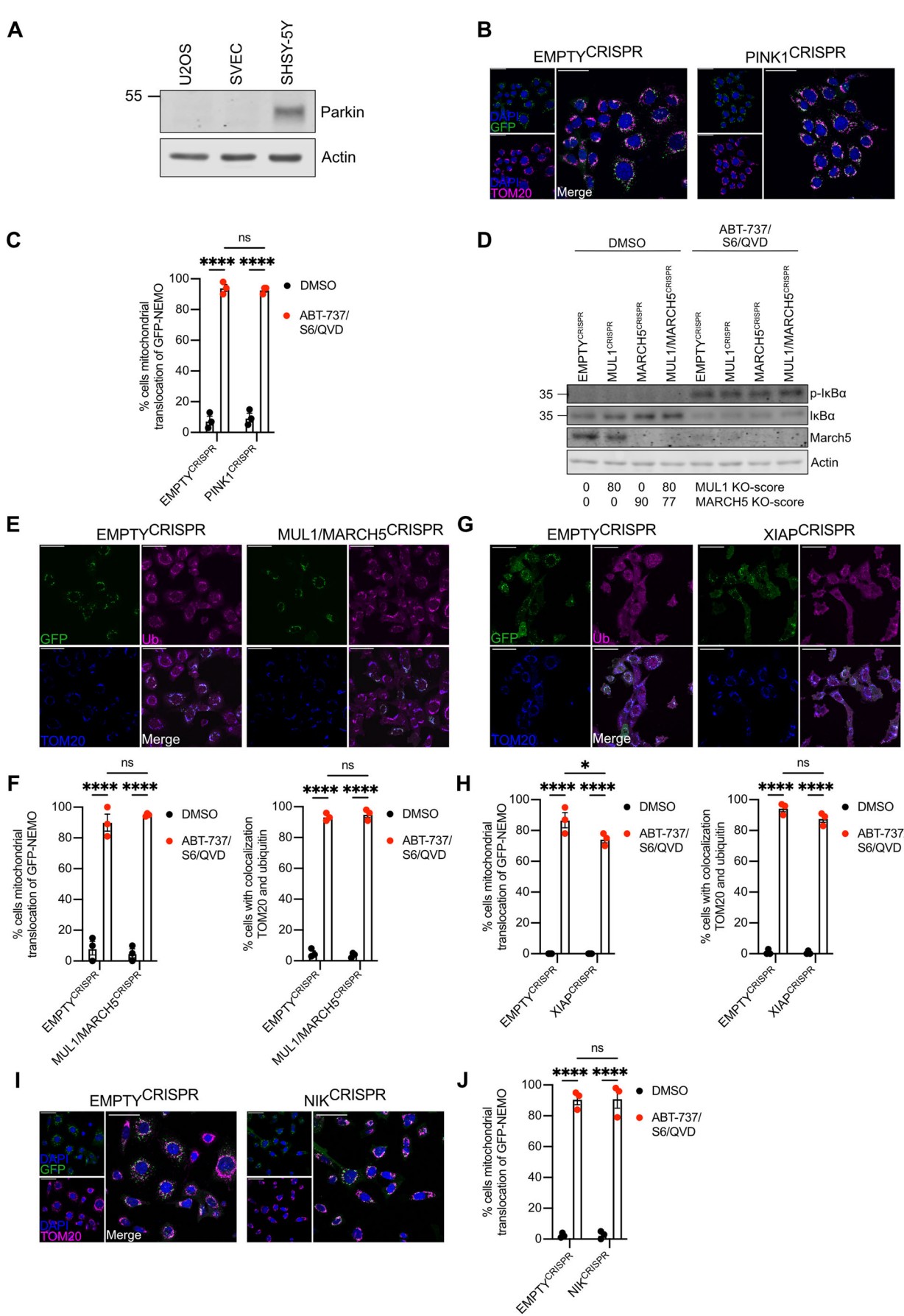

**Figure 5. Ubiquitylation-induced inflammation after MOMP is independent of established mitochondrial E3 ligases.**

(A) Lysates of U2OS, SVEC4-10 and SHSHY-5Y cells were blotted for Parkin and Actin. (B) SVEC4-10 EMPTY[CRISPR] and PINK1[CRISPR] cells expressing GFP-NEMO were treated for 1 h with 10 µM ABT-737, 10 µM S63845 and 30 µM Q-VD-OPh. Cells were immunostained for mitochondrial TOM20 and DAPI. Images are representative for three independent experiments with a scale bar of 50 µm. (C) Quantification of (B) showing the percentage of cells with mitochondrial translocation of GFP-NEMO. (D) SVEC4-10 EMPTY[CRISPR], MUL1[CRISPR], MARCH5[CRISPR] and MUL1/MARCH5[CRISPR] treated for 3 h with 10 µM ABT-737, 10 µM S63845 and 30 µM Q-VD-OPh. Lysates were blotted for p-IκBα, IκBα, MARCH5 and Actin. Blots are representative of three independent experiments. KO-scores of MUL1 and MARCH5 are calculated via ICE analysis. (E) SVEC4-10 EMPTY[CRISPR] and MUL1/MARCH5[CRISPR] cells expressing GFP-NEMO treated for 1 h with 10 µM ABT-737, 10 µM S63845 and 30 µM Q-VD-OPh. Cells were immunostained for ubiquitin (UBCJ2) and mitochondrial TOM20. Images are representative for three independent experiments with a scale bar of 50 µm. (F) Quantification of (E) showing the percentage of cells with mitochondrial localisation of GFP-NEMO and ubiquitin. (G) SVEC4-10 EMPTY[CRISPR] and XIAP[CRISPR] cells expressing GFP-NEMO were treated with 10 µM ABT-737, 10 µM S63845 and 30 µM Q-VD-OPh for 1 h. Cells were immunostained for ubiquitin (FK2) and mitochondrial TOM20. Images are representative for three independent experiments with a scale bar of 50 µm. (H) Quantification of (G) showing the percentage of cells with mitochondrial localisation of GFP-NEMO and ubiquitin. (I) SVEC4-10 EMPTY[CRISPR] and NIK[CRISPR] cells expressing GFP-NEMO were treated with 10 µM ABT-737, 10 µM S63845 and 30 µM Q-VD-OPh for 1 h. Images are immunostained with mitochondrial TOM20 and DAPI. Images are representative for three independent experiments with a scale bar of 50 µm. (J) Quantification of (I) showing the percentage of cells with mitochondrial localisation of GFP-NEMO. Data information: (C, F, H, J) graphs display mean values ± s.e.m. (error bars) of $n = 3$ independent experiments. Statistics were performed using two-way ANOVA with Tukey correction. ****$P < 0.0001$. Source data are available online for this figure.

## Generation of GFP-NEMOD311N and GFP-NEMOΔZF

GFP-NEMOD311N and GFP-NEMOΔZF were cloned into a pBABE-puro vector using EcoRI and BamHI restriction sites. GFP-NEMOD311N was cloned into the pBABE vector using Gibson assembly. NEMOD311N was obtained by PCR of pGEX-NEMOD311N (Addgene #11968). GFP was obtained by PCR of a GFP-containing plasmid. GFP-NEMOΔZF was obtained by PCR of the M6P-GFP-NEMO plasmid (gifted by Felix Randow), thereby removing the last 25 amino acids of wild-type NEMO.

GFP 5'-tctaggcgccggccggatccATGGTGAGCAAGGGCGAG-3'
GFP 3'-cagaaccaccaccaccCTTGTACAGCTCGTCCATGC-5'
NEMOD311N
5'-ctgtacaagggtggtggtggttctggtggtggtggttctAATAGGCACCTCTGGAAG-3'
NEMOD311N 3'-accactgtgctggcgaattcCTACTCAATGCACTCCATG-5'
GFP-NEMOΔZF 5'-TAAGCA GGATTCATGGTGAGCAAGGGCGAGGAG-3'
GFP-NEMOΔZF 3'-TGCTTA GAATTC CTAGTCAGGTGGCTCCTCGGGGG-5'

## ICE analysis for CRISPR

Genomic DNA was isolated from the empty vector and CRISPR cells. A PCR reaction for the CRISPR'ed region was set up using Phusion DNA polymerase according to the manufacturer's instructions. The reactions were run on 2% agarose gel and bands of the correct size were isolated and purified using the GeneJET Gel Extraction kit. Samples were sequenced by Eurofins genomics and analysed using ICE software by Synthego.

Mouse *March5* 5'-TCCTGGCCTGAAGGGTAGGGGA-3'
Mouse *March5* 3'-CCTCTTCCTTCCCCCACCCCAA-5'
Mouse *Mul1* 5'-GGGTCGCAGGTGATTTCGAGGC-3'
Mouse *Mul1* 3'-CACGTTGGAATCACCCCTGCCT-5'
Mouse *Pink1* 5'-TGTTGTTGTCCCAGACGTTTGT-3'
Mouse *Pink1* 3'-TAAATTGCCCAATCACGGCTCA-5'

## Knockdown using siRNA

SVEC4-10 cells were transfected with 20 nM siGENOME *Nemo* SMARTpool (Horizon Discovery #M-040796-01-0005) or siGENOME non-targeting control (Dharmacon #D0012061305) using lipofectamine RNAiMAX (Invitrogen #1377075). Experiments were performed 48 h after transfection.

## RT-qPCR

RNA was isolated using the GeneJET RNA isolation kit (Thermo Fisher Scientific #K0732) according to the manufacturer's instructions. Genomic DNA was digested using an on-column DNase step (Sigma-Aldrich #04716728001) for 15 min. RNA was converted into cDNA using the High Capacity cDNA Reverse Transcriptase kit (Thermo Fisher Scientific #43-688-13) according to the manufacturer's instructions. cDNA was synthesised according to the following steps: 25 °C for 10 min, 37 °C for 120 min and 85 °C for 5 min.

RT-qPCR was performed by using the Brilliant III SYBR® Green QPCR Master Mix (Agilent #600882) or DyNAmo HS SYBR Green (Thermo Scientific #F410L) and the QuantStudio 3. The following RT-qPCR cycling parameters were used: initial denaturation on 95 °C for 10 min, 40 cycles of 95 °C for 20 s, 57 °C for 30 s and 72 °C for 30 s, finished by a dissociation step 65–95 °C (0.5 °C/s). Samples were run in technical triplicates. Fold change expression was determined using the $2^{-\Delta\Delta CT}$ method.

### cDNA

Mouse *Actin* 5'-CTAAGGCCAACCGTGAAAAG-3'
Mouse *Actin* 3'-ACCAGAGGCATACAGGGACA-5'
Mouse *Ccl5* 5'-CTGCTGCTTTGCCTACCTCT-3'
Mouse *Ccl5* 3'-CGAGTGACAAACACGACTGC-5'
Mouse *Kc* 5'-GGCTGGGATTCACCTCAAGAA-3'
Mouse *Kc* 3'-GAGTGTGGCTATGACTTCGGTT-5'
Mouse *Tnf-α* 5'-GTCCCCAAAGGGATGAGAAG-3'
Mouse *Tnf-α* 3'-CACTTGGTGGTTTGCTACGAC-5'

### DNA

Human *CYTB* 5'-GCCTGCCTGATCCTCCAAAT-3'
Human *CYTB* 3'-AAGGTAGCGGATGATTCAGCC-5'
Human *GAPDH* 5'-TGGGGACTGGCTTTCCCATAA-3'
Human *GAPDH* 5'-CACATCACCCCTCTACCTCC-3'

## Western blotting

Cells were lysed in RIPA buffer (10 mM Tris-HCl (pH 7.4), 150 mM NaCl, 1.2 mM EDTA, 1% Triton X-100 and 0.1% SDS

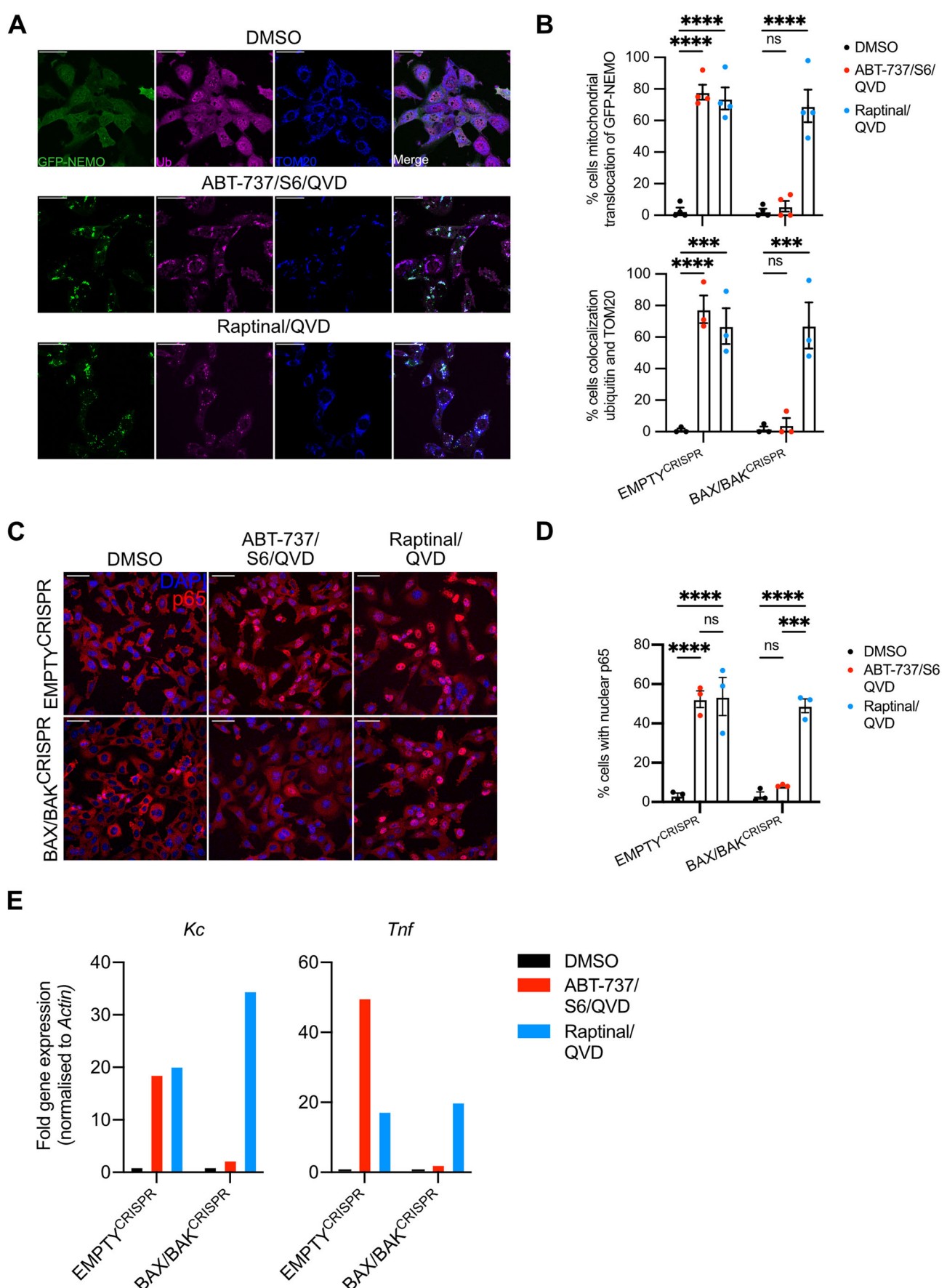

Figure 6.  Mitochondrial ubiquitylation and inflammation occur upon loss of mitochondrial outer membrane integrity.

(A) SVEC4-10 cells expressing GFP-NEMO were treated for 3 h with 10 µM ABT-737, 10 µM S63845 and 30 µM Q-VD-OPh or 2.5 µM raptinal and 30 µM Q-VD-OPh. Cells were immunostained for ubiquitin (FK2) and mitochondrial TOM20. Images are representative of three independent experiments displayed with a 50 µm scale bar. (B) SVEC EMPTY$^{CRISPR}$ and BAX/BAK$^{CRISPR}$ cells were treated for 3 h with 10 µM ABT-737, 10 µM S63845 and 30 µM Q-VD-OPh or 2.5 µM raptinal and 30 µM Q-VD-OPh. Graphs show the percentage of cells with mitochondrial localisation of GFP-NEMO and ubiquitin. (C) SVEC4-10 EMPTY$^{CRISPR}$ and BAX/BAK$^{CRISPR}$ cells were treated for 3 h with 10 µM ABT-737, 10 µM S63845 and 30 µM Q-VD-OPh or 2.5 µM raptinal and 30 µM Q-VD-OPh. Cells were immunostained stained for p65 and DAPI. Images are representative of three independent experiments. Scale bar is 50 µm. (D) Quantification of (C) showing the percentage of cells with nuclear translocation of p65. (E) SVEC4-10 cells treated for 3 h with 10 µM ABT-737, 10 µM S63856 and 30 µM Q-VD-OPh or 2.5 µM raptinal and 30 µM Q-VD-OPh. Expression of *Kc*, *Tnf* and *Actin* were validated using RT-qPCR, graphs are representative for three independent experiments. Data information: (B, D) graphs display mean values ± s.e.m. (error bars) of $n = 3$ independent experiments. Statistics were performed using two-way ANOVA with Dunnett correction. ****$P < 0.0001$. Source data are available online for this figure.

supplemented with cOmplete protease inhibitors) and proteins were isolated by maximal centrifugation (15,000 rpm) for 10 min. Lysates were loaded on 8, 10 or 12% gels and transferred onto the nitrocellulose membranes. The membranes were blocked with 5% milk or BSA in TBS for 1 h followed by overnight incubation of 1:1000 dilution of primary antibodies in 5% milk or BSA in TBS-T. The next day, membranes were incubated with a 1:10,000 dilution of secondary antibodies for 1 h and imaged on the Li-cor CLx. Primary antibodies used are actin (Sigma #A4700), ATG5 (CST #8540), ATG7 (CST #8558), BAK (CST #12105), BAX (CST #2772), COXIV (CST #11967), FK2 (ENZO #BML-PW8810-0100), GAPDH (CST #2118), HSP60 (Santa Cruz #sc-13115), p-IκBα (CST #2859), IκBα (CST #4814), K48-ubiquitin (CST #8081), K63-ubiquitin (Merck #05-1308), LC3 (CST #2775), MARCH5 (EMD Millipore #06-1036), Membrane Integrity Antibody cocktail (Abcam #ab110414), NEDD8 (Abcam #AB81264), NEMO (Abcam #178872), NIK (CST #4994), Parkin (Santa Cruz #sc-32282), SMAC (Abcam #AB32023), TOMM20 (Proteintech #11082-1-AP), UBCJ2 (ENZO #ENZ-ABS840-0100), and XIAP (BD #610716). Secondary antibodies used are goat anti-rabbit IgG (H + L) Alexa Fluor Plus 800 (Invitrogen #A32735), goat anti-mouse IgG (H + L) Alexa Fluor 680 (Invitrogen #A21057) and goat anti-mouse IgG (H + L) Dylight 800 (Invitrogen #SA535521).

## Mitochondrial isolation using digitonin

Cells were lysed in digitonin lysis buffer (0.25 M sucrose, 700 mM Tris-HCl pH 8 and 100 µg/mL digitonin) for 10 min on ice. The mitochondrial fraction was pelleted at 3000 × *g* for 5 min. The supernatant was stored as the non-mitochondrial fraction, the pellet was resuspended in RIPA lysis buffer and stored on ice for 20 min followed by centrifugation for 10 min at maximum speed (15,000 rpm). The supernatant was taken as a mitochondrial fraction.

## Mitochondrial isolation using Dounce homogeniser

Cells were resuspended in mitochondrial isolation buffer (200 mM mannitol, 70 mM sucrose, 10 mM HEPES, 1 mM EGTA, pH 7.0, cOmplete protease inhibitor). After resuspension cells were homogenised using the Dounce tissue grinder by performing 50 strokes up/down manually and centrifuged at 2000 rpm for 5 min. The supernatant was collected and the pellet was resuspended in mitochondrial isolation buffer and spun down as previously described. Supernatant from both spins were combined and spun down at 9000 rpm for 5 min. The supernatant was kept as non-mitochondrial fraction. The pellet was resuspended in RIPA buffer and placed on ice for 20 min followed by centrifugation at maximum speed (15,000 rpm) for 10 min. The supernatant was kept as a mitochondrial fraction.

## Immunofluorescent staining

Cells were fixed using 4% PFA for 15 min, followed by a 15 min permeabilization step using 0.2% Triton X-100. Samples were blocked using 2% BSA in PBS for 1 h and incubated with primary antibody in 2% BSA overnight. The following day, samples were incubated with a secondary antibody in 2% BSA. Primary antibodies used are COXIV (CST #11967 and #4850), cytochrome *c* (BD #556432), FK2 (ENZO #BML-PW8810-0100), HSP60 (Santa Cruz #sc-13115), K63-ubiquitin (Merck #05-1308), p65 (CST #8242), TOM20 (CST #42406 and Proteintech #11082-1-AP) and UBCJ2 (ENZO #ENZ-ABS840-0100). Secondary antibodies used are Alexa Fluor 488 goat anti-rabbit IgG (H + L) (Invitrogen #A11034), Alexa Fluor 488 goat anti-mouse IgG (H + L) (Invitrogen #A11029), Alexa Fluor 568 goat anti-rabbit IgG (H + L) (Invitrogen #A11011), Alexa Fluor 568 goat anti-mouse IgG (H + L) (Invitrogen #A11004), Alexa Fluor 647 goat anti-rabbit IgG (H + L) (Invitrogen #A21245) and Alexa Fluor 647 goat anti-mouse IgG (H + L) (Invitrogen #A21236). Coverslips were mounted using Vectashield or ProLong™ Glass Antifade Mountant (Invitrogen #P36980) with or without DAPI.

## Confocal microscopy

Fixed samples were imaged using the Nikon A1R confocal microscope using all four lasers (405 nm, 488 nm, 561 nm and 638 nm) and images are acquired using sequential scanning. For p65 staining the 40× NA 1.30 oil-immersion objective was used, while the 60 × 1.40 NA oil-immersion objective was used to determine ubiquitin, GFP-NEMO and YFP-Parkin puncta. Images were analysed using ImageJ version 2.1.0/1.53c and cells with mitochondrial ubiquitin or GFP-NEMO translocation to mitochondria was counted using the cell counter plugin. Specifically, cells of at least three images per condition from three independent experiments were counted using the Cell Count plugin in Fiji. In addition, the plugin was used to quantify number of cells with overlapping ubiquitin stain or GFP-NEMO with the mitochondrial stain (COXIV, TOMM20). The same method was used for the quantification of nuclear p65. Images may be displayed using pseudocolours.

## Live-cell imaging and time-lapse microscopy

Cells for live-cell imaging and time-lapse microscopy were seeded into µ-Slides (ibidi #80807) culture chambers or 35-mm dishes (MaTek #P35G-1.5-20-C). Treatment of cells with DMSO or S6/ABT-737/Q-VD-Oph was initiated as indicated in the Fig legends. MitoTracker Green (200 nM), PkMito DeepRed (1:2000) and Hoechst (1:2000) were added to cells 1 h, 30 min and 15 min prior to imaging and cells washed in the case of MitoTracker Green/Hoechst staining. Images were taken using a

Zeiss LSM 880 point-scanning confocal microscope on an inverted Zeiss Axio Observer.Z1 stand and 37 °C temperature as well as 5% $CO_2$ maintained using a stage top- and cage incubator (P S1, XLmulsti S1, Pecon). To avoid bleed-through, images were acquired sequentially using a 63×/1.4 Plan-Apochromat lens with immersion oil using the 488, 561, and 640 nm laser lines, with a 1× zoom resulting in a pixel size of at least 1024 × 1024. All images were acquired using the software Zen LSM 2.1 Black (Zeiss). For image analysis of time-lapse images Fiji/ImageJ version 2.9.0/1.53t was used (Schindelin et al, 2012). For the quantification of GFP-Nemo puncta on mitochondria, each timepoint was analysed using the plugin "trainable Weka segmentation" (Arganda-Carreras et al, 2017) previously trained to identify and mask mitochondria (using PkMito DeepRed). In a second step GFP-NEMO puncta on the mitochondrial mask were identified and counted using the function find maxima. Results were plotted using GraphPad Prism.

## Cell death assays using Incucyte

Cell death assays were performed using Incucyte ZOOM from Sartorius. Cell death was measured by Sytox Green inclusion (Thermo Fisher Scientific #S7020). Images were taken every hour with a 10× objective. Starting confluency was used for normalisation.

## Isolation of peptides containing ubiquitin remnants

Peptides containing ubiquitin remnant motifs were isolated using the PTMScan® Ubiquitin Remnant Motif (K--GG) Kit (CST #5562) according to the manufacturer's instructions. Isolation of ubiquitin remnants was performed on four independent repeats for both conditions (4.4 mg protein per sample). Cellular localisation of proteins was determined using Uniprot and Proteinatlas. Mitochondrial localisation was determined using MitoCarta 3.0. GO enrichment analysis was performed using PANTHER classification system.

## Mitosox and RHOD2-AM measurements by Flow cytometry

For measurement of mitochondrial reactive oxygen species (mtROS) SVEC4-10 cells were treated with 500 nM MitoSOX™ Red (Thermo Fisher Scientific, M36008) and 3 μM erastin, 1 and 5 μM antimycin, or 1 μM rotenone, simultaneously, for 2 h prior to harvesting. For the measurement of mitochondrial calcium, SVEC4-10 cells were treated with mitochondrial toxins shown above. RHOD2-AM (Thermo Fisher Scientific, #R1244) was prepared and added to cells 30 min prior to harvest as per the manufacturers' instructions. Harvest was conducted by trypsinisation and quenching with fully supplemented media. Fluorescence intensity for both reporters was recorded by Attune NxT flow cytometer as per the manufacturers' instructions.

## Mass spectrometry

Peptides were separated by nanoscale C18 reverse-phase liquid chromatography using an EASY-nLC II 1200 (Thermo Scientific) coupled to an Orbitrap Fusion Lumos mass spectrometer (Thermo Scientific). Elution was performed at a flow rate of 300 nL/min using a binary gradient, into a 50 cm fused silica emitter (New Objective) packed in-house with ReproSil-Pur C18-AQ, 1.9-μm resin (Dr Maisch GmbH), for a total duration of 135 min. Packed emitter was kept at 50 °C by column oven (Sonation) integration into the nanoelectrospray ion source (Thermo Scientific). Eluting peptides were electrosprayed into the mass spectrometer using a nanoelectrospray ion source. To decrease air contaminants signal level an Active Background Ion Reduction Device (EDI Source Solutions) was used. Data acquisition was performed using Xcalibur software (Thermo Scientific). A full scan over mass range of 350–1400 $m/z$ was acquired at 120,000 resolution at 200 $m/z$. Higher energy collision dissociation fragmentation was performed on the 15 most intense ions, and peptide fragments generated were analysed in the Orbitrap at 15,000 resolution.

The MS Raw data were processed using MaxQuant software version 1.6.3.3 and searched with Andromeda search engine (Cox et al, 2011) querying SwissProt (Consortium 2019) Mus musculus (20/06/2016; 57,258 entries). First and main searches were performed with precursor mass tolerances of 20 ppm and 4.5 ppm, respectively, and MS/MS tolerance of 20 ppm. The minimum peptide length was set to six amino acids and specificity for trypsin cleavage was required. Methionine oxidation, N-terminal acetylation and di-Gly-lysine were specified as variable modifications, whereas cysteine carbamidomethylation was set as fixed modification. The peptide, protein, and site false discovery rate (FDR) was set to 1%. All MaxQuant outputs were analysed with Perseus software version 1.6.2.3 (Tyanova et al, 2016).

The MaxQuant output GlyGly (K)sites.txt file was use for quantification of Ubiquitylated peptides. From the GlyGly (K)Sites.txt file, Reverse and Potential Contaminant flagged peptides (defined as MaxQuant output) were removed. To determine significantly changing ubiquitylated peptides a Student $t$ test with a 1% FDR (permutation-based) was applied using the peptide intensities included in the GlyGly (K) sites table. Missing values were input separately for each column (width 0.3, downshift 1.4). Only ubiquitylated peptides having: "score diff" greater than 5, a localisation probability higher than 0.75, and are robustly quantified in three out of four replicate experiments were included in the analysis.

## Statistics

Statistics was performed using Prism 9. All data represent mean ± standard error of the mean (s.e.m.) unless indicated differently.

$*P < 0.05$, $**P < 0.01$, $***P < 0.001$, $****P < 0.0001$.

# Data availability

The raw files and the MaxQuant search results files have been deposited to the ProteomeXchange Consortium (Deutsch et al, 2020) via the PRIDE partner repository (Perez-Riverol et al, 2022) with the dataset identifier PXD040192. Data are available via ProteomeXchange (https://www.ebi.ac.uk/pride/archive/projects/PXD040192) with identifier PXD040192.

# Peer review information

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

## Acknowledgements

This research was supported by funding from the Cancer Research UK [(DRCNPG-Jun22\100011 and A20145; SWGT)(A29256; DTH)((DRCNPG-Jun22\100007 and EDDPGM-Nov21\100001; DJM). Stand Up to Cancer campaign for CRUK A29800 to SZ; A31287 to CRUK Beatson Institute, A18076 to CRUK Glasgow Centre, A17196 to CRUK Beatson Institute Advanced Technology Facilities. RH is supported by a fellowship from the Swiss National Science Foundation and Novartis Foundation for Medical-Biological Research. HW is supported by an Alexander von Humboldt Foundation Professorship Award, a Cancer Research UK Programme Grant (A27323), a Wellcome Trust Investigator Award (214342/Z/18/Z), a Medical Research Council Grant (MR/S00811X/1) and three collaborative research centre grants funded by Deutsche Forschungsgemeinschaft (DFG, German Research Foundation): SFB1399, Project C06; SFB1530-455784452, Project A03; and SFB1403-414786233. MG-H is supported by the LEO Foundation and The Novo Nordisk Foundation (NNF20OC0059392). We thank Beatson Advanced Imaging Resource (BAIR) specifically Ryan Corbyn and Claire Mitchell for their help with image analysis and automation of image analysis. We thank Asma Ahmed and Catherine Winchester for critical reading of the manuscript.

## Author contributions

**Esmee Vringer**: Conceptualisation; Data curation; Formal analysis; Investigation; Writing—original draft; Writing—review and editing. **Rosalie Heilig**: Data curation; Formal analysis; Validation; Investigation; Writing—review and editing. **Joel S Riley**: Investigation; Methodology; Writing—review and editing. **Annabel Black**: Investigation; Writing—review and editing. **Catherine Cloix**: Investigation; Methodology; Writing—review and editing. **George Skalka**: Data curation; Formal analysis; Investigation. **Alfredo E Montes-Gómez**: Formal analysis. **Aurore Aguado**: Investigation. **Sergio Lilla**: Investigation; Methodology; Writing—review and editing. **Henning Walczak**: Resources; Methodology; Writing—review and editing. **Mads Gyrd-Hansen**: Methodology; Writing—review and editing. **Daniel J Murphy**: Funding acquisition. **Danny T Huang**: Data curation; Methodology; Writing—original draft; Writing—review and editing. **Sara Zanivan**: Supervision; Writing—review and editing. **Stephen WG Tait**: Conceptualisation; Formal analysis; Supervision; Funding acquisition; Writing—original draft; Writing—review and editing.

## Disclosure and competing interests statement

SWGT consults for Exo Therapeutics. DTH consults for Triana Biomedicines. Salary support for GS was provided by a collaborative agreement with the Merck Group and Cancer Research Horizons UK. DJM has additionally received support from Puma Biotechnology and ORIC Pharmaceuticals. The remaining authors declare no competing interests.

# Expanded View Figures

**Figure EV1. Ubiquitylation of mitochondria is dependent on MOMP by BAX/BAK pores, but independent of caspase activity.**

(**A**) U2OS EMPTY^CRISPR and BAX/BAK^CRISPR cells were treated with 10 µM ABT-737, 2 µM S63845 with or without 20 µM Q-VD-OPh. Cell death was determined using Sytox Green inclusion normalised to starting confluence. Graph is representative of three independent experiments and displays mean values ± s.e.m. (error bars) of technical triplicates. (**B**) Lysates from U2OS EMPTY^CRISPR and BAX/BAK^CRISPR cells were blotted for BAX, BAK and Actin. (**C**) SVEC4-10 EMPTY^CRISPR and BAX/BAK^CRISPR cells were treated for 1 h with 10 µM ABT-737, 10 µM S63845 and 30 µM Q-VD-OPh. Mitochondria were isolated using digitonin fractionation buffer and blotted for ubiquitin (UBCJ2), BAX, BAK, HSP60 and Actin. (**D**) SVEC4-10 cells were treated for 1 h with 10 µM ABT-737, 10 µM S63845 with or without 30 µM Q-VD-OPh. Mitochondria were isolated using digitonin fractionation buffer and lysates were blotted for ubiquitin (UBCJ2), HSP60 and actin. (**E**) U2OS cells were treated for 7 h with 10 µM ABT-737, 2 µM S63845 and 20 µM Q-VD-OPh with or without the addition of 10 µM MG-132. Lysates were blotted for ubiquitin (UBCJ2) and GAPDH. (**F**) U2OS cells were treated for 24 h with 10 µM ABT-737, 2 µM S63845 and 20 µM Q-VD-OPh with or without the addition of 10 µM MG-132. Mitochondrial depletion was assessed by blotting for several mitochondrial proteins and GAPDH. (**G**) U2OS cells were treated for 7 h with 10 µM ABT-737, 2 µM S63845 and 20 µM Q-VD-OPh with or without the addition of 1 µM TAK-243. Lysates were blotted for ubiquitin (UBCJ2) and GAPDH. (**H**) U2OS cells were treated for 24 h with 10 µM ABT-737, 2 µM S63845 and 20 µM Q-VD-OPh with or without the addition of 1 µM TAK-243. Mitochondrial depletion was assessed by blotting for several mitochondrial proteins and GAPDH. Data information: (**C, D, E, F, G, H**) blots are representative of three independent experiments.

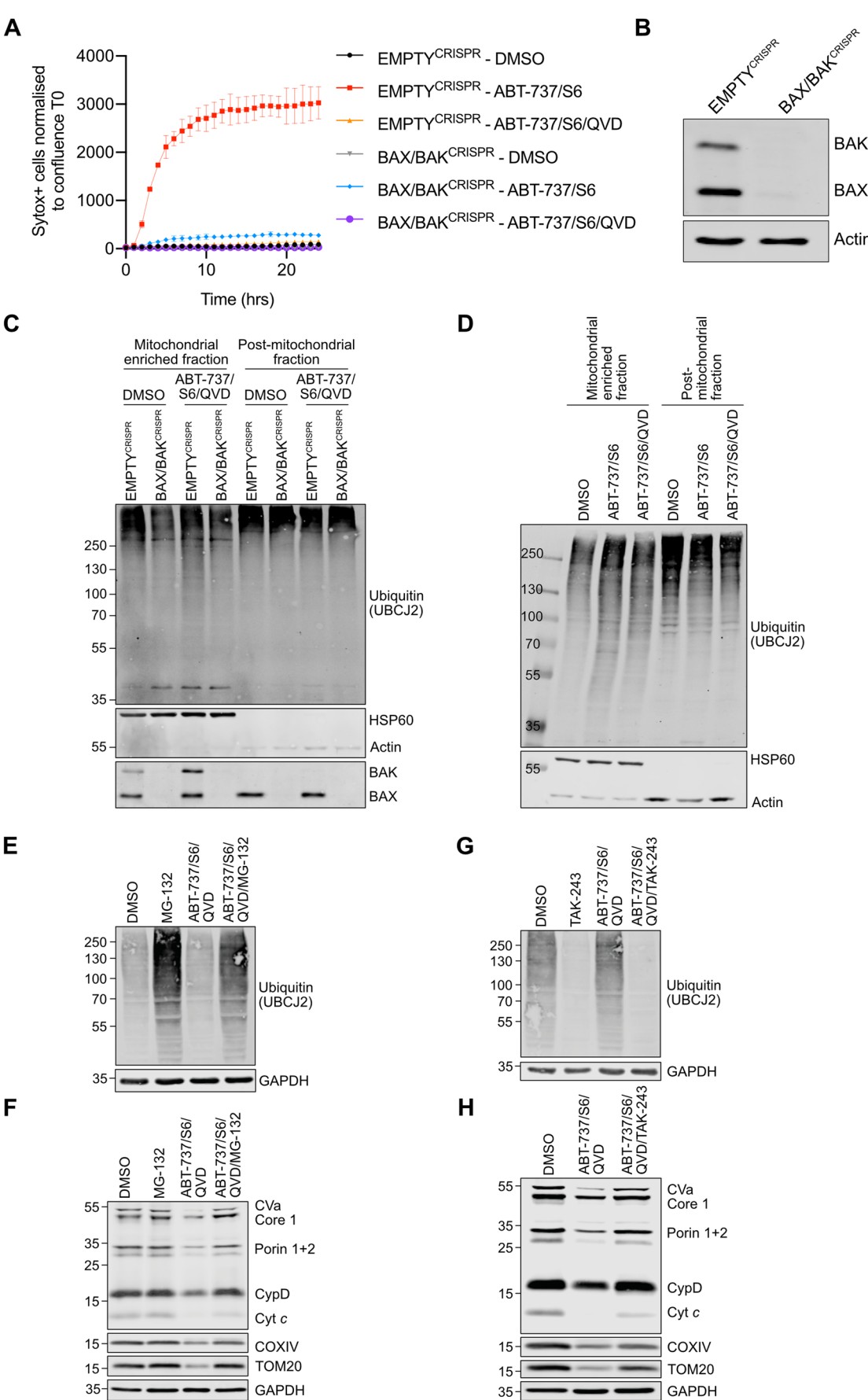

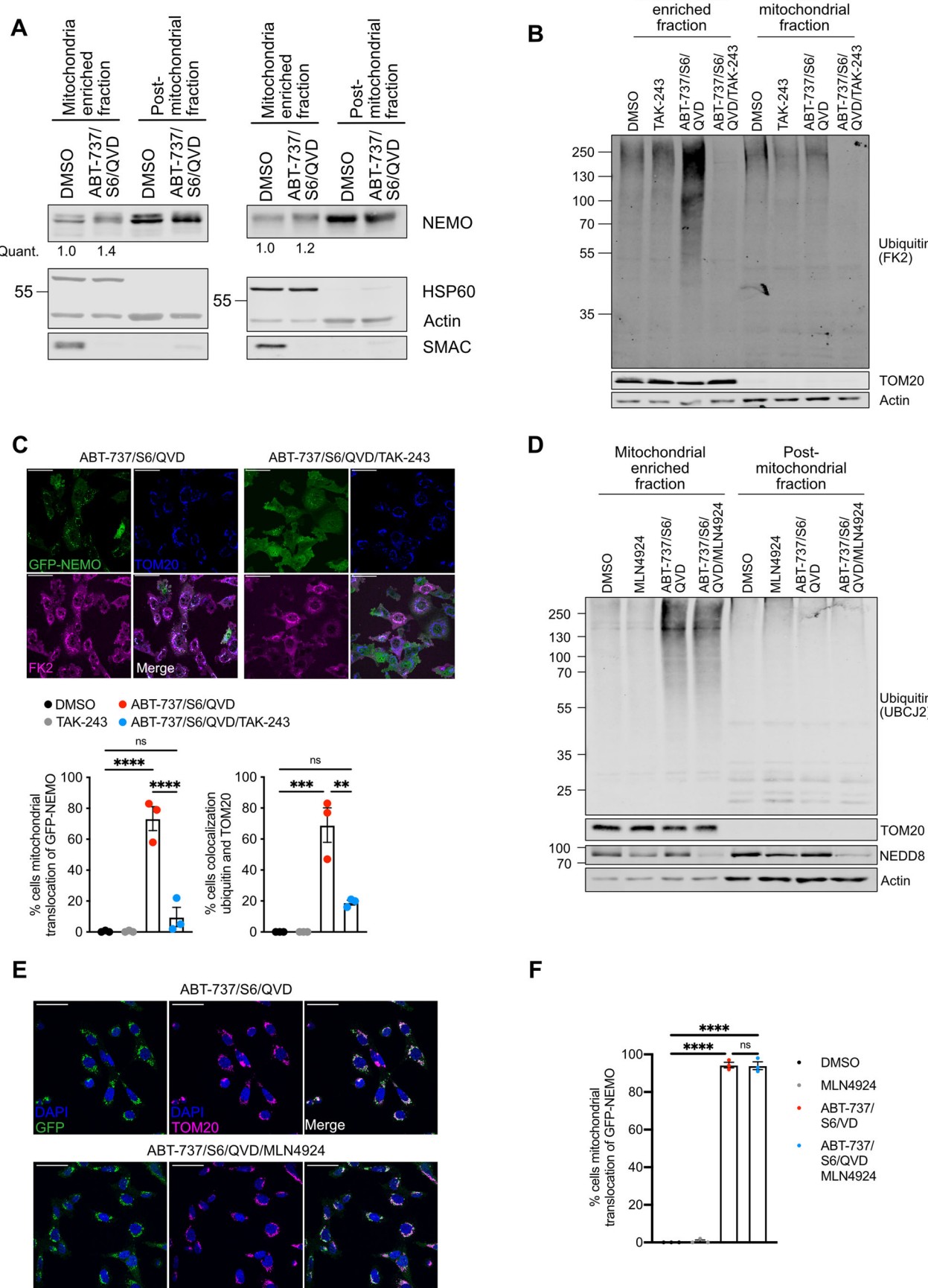

◀ **Figure EV2.   Mitochondrial ubiquitylation and GFP-NEMO translocation can be blocked by E1 inhibition and is independent of neddylation.**

(A) SVEC4-10 cells treated for 3 h with 10 µM ABT-737, 10 µM S63845 and 30 µM Q-VD-OPh. Mitochondria were isolated using Dounce homogeniser and cellular fractions were probed with relevant antibodies. Mitochondrial localised NEMO was quantified normalising to mitochondrial content defined by HSP60 signal. (B) SVEC4-10 cells pre-treated with 2 µM TAK-243 for 1 h followed by additional 1 h treatment with 10 µM ABT-737, 10 µM S63845 and 30 µM Q-VD-OPh with or without 2 µM TAK-243. Blots are representative for four independent experiments. (C) Upper: SVEC4-10 cells expressing GFP-NEMO were pre-treated for 1 h with 2 µM TAK-243 followed by 1 h treatment of 10 µM ABT-737, 10 µM S63845, 30 µM Q-VD-OPh with or without 2 µM TAK-243. Cells were immunostained for TOM20 and ubiquitin (FK2). Scale bar is 50 µm and images are representative for three independent experiments. Lower: quantification showing the percentage of cells with mitochondrial localised GFP-NEMO and ubiquitin puncta. (D) SVEC4-10 cells pre-treated with 1 µM MLN4924 (NAE inhibitor) for 1 h followed by additional 1 h treatment with 10 µM ABT-737, 10 µM S63845 and 30 µM Q-VD-OPh with or without 1 µM MLN4924. Blots are representative for two independent experiments. (E) SVEC4-10 cells expressing GFP-NEMO pre-treated with 1 µM MLN4924 for 1 h followed by additional 1 h treatment with 10 µM ABT-737, 10 µM S63845 and 30 µM Q-VD-OPh with or without 1 µM MLN4924. Cells were immunostained for mitochondrial TOM20 and DAPI. Images are representative for three independent experiments and are shown with a 50 µm scale bar.

(F) Quantification of (E) showing the percentage of cells with mitochondrial translocation of GFP-NEMO. Data information: graphs in (C, F) display mean values ± s.e.m. (error bars) of $n = 3$ independent experiments. Statistics were performed using two-way ANOVA with Tukey correction. ****$P < 0.0001$.

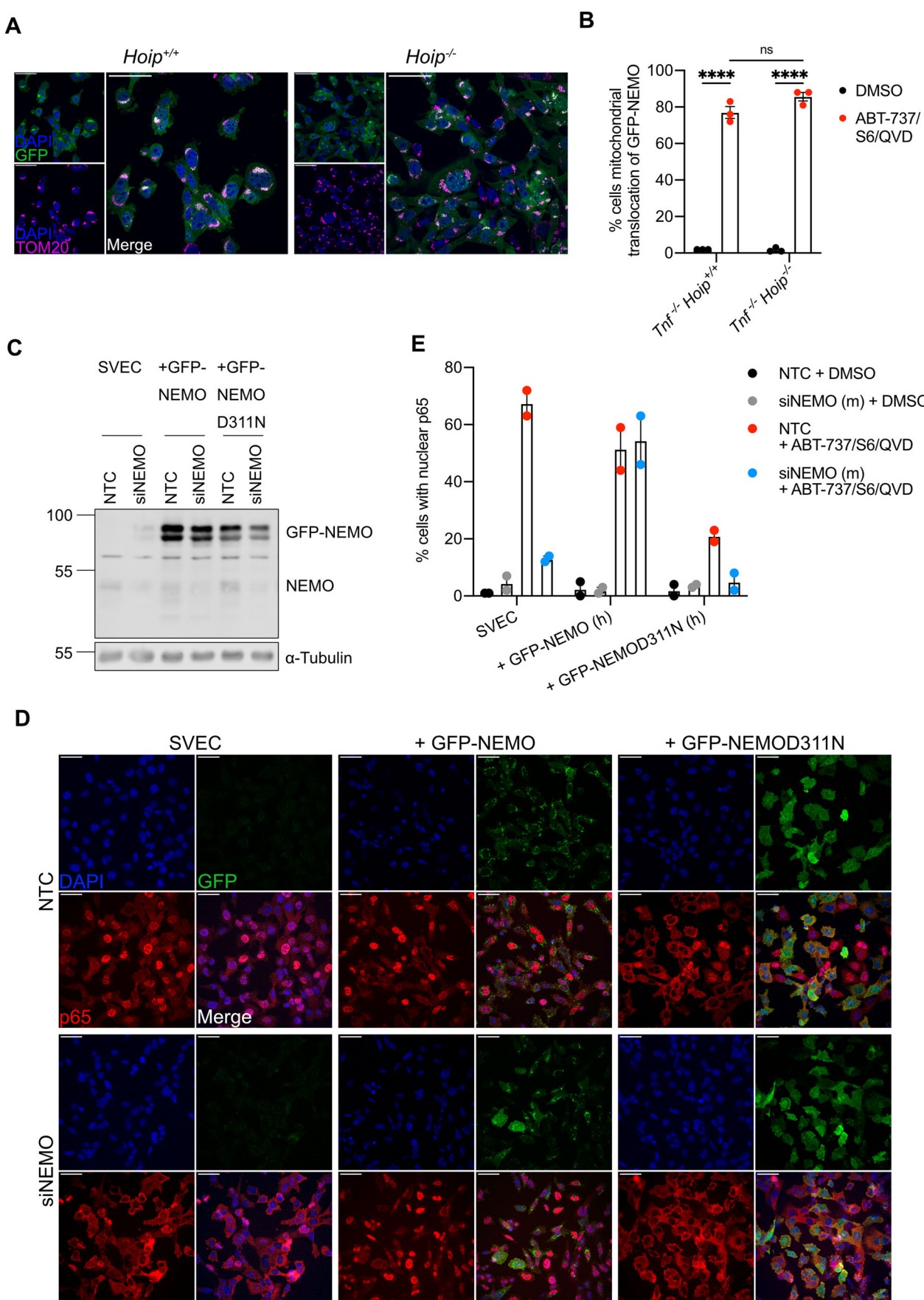

**Figure EV3.  Loss of NEMO cannot be rescued during CICD by expressing non-ubiquitin-binding mutants of NEMO.**

(A) MEF Tnf$^{-/-}$/Hoip$^{+/+}$ and Tnf$^{-/-}$/Hoip$^{-/-}$ expressing GFP-NEMO were treated for 3 h with 10 µM ABT-737, 5 µM S63845 and 30 µM Q-VD-OPh. Cells were immunostained for mitochondrial TOM20 and DAPI. Images are representative of three independent experiments. (B) Quantification of A showing the percentage of cells with mitochondrial translocation of GFP-NEMO. Graph displays mean values ± s.e.m. (error bars) of $n = 3$ independent experiments. Statistical analysis was performed using two-way ANOVA with Tukey correction. (C) Validation of SVEC4-10, SVEC4-10 GFP-NEMO and SVEC4-10 GFP-D311N cells transfected with NTC or siNEMO. Lysates were blotted for NEMO and α-tubulin. (D) SVEC4-10, SVEC4-10 GFP-NEMO and SVEC4-10 GFP-D311N cells transfected with NTC or siNEMO were treated for 1 h with 10 µM ABT-737, 10 µM S63845 and 30 µM Q-VD-OPh. Cells were immunostained for p65 and DAPI. Images are representative of two independent experiments. (E) Quantification of (D) showing the percentage of cells with nuclear translocation of p65. Graph displays mean values ± s.e.m. (error bars) of $n = 2$ independent experiments. ****$P < 0.0001$.

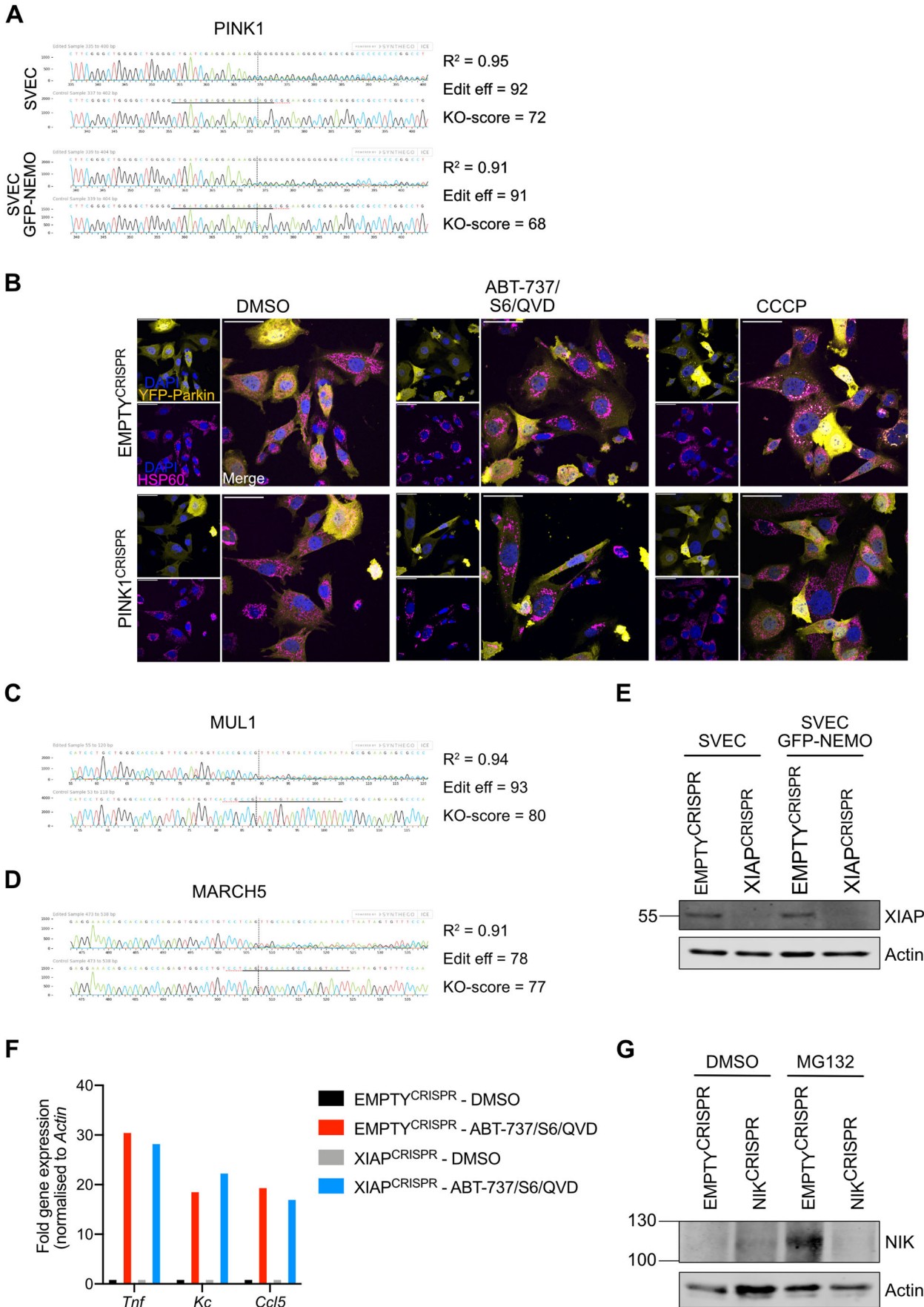

◀   **Figure EV4.   Validation of PINK1^CRISPR, NIK^CRISPR, MUL1/MARCH5^CRISPR and XIAP^CRISPR knockout cell lines.**

(**A**) Validation of PINK1 knockout in SVEC4-10 cells with or without GFP-NEMO expression using genomic PCR and ICE (inference of CRISPR edits) analysis. (**B**) SVEC4-10 EMPTY^CRISPR and PINK1^CRISPR cells expressing YFP-Parkin were treated for 1 h with 10 µM ABT-737, 10 µM S63845 and 30 µM Q-VD-OPh or for 3 h with 10 µM CCCP. Mitochondria were immunostained with HSP60 and DAPI. Images are representative of two independent experiments and displayed with 50 µm scale bar. (**C**) Validation of MUL1 knockout in SVEC4-10 MUL1/MARCH5^CRISPR cells using genomic PCR and ICE analysis. (**D**) Validation of MARCH5 knockout in SVEC4-10 MUL1/MARCH5^CRISPR cells using genomic PCR and ICE analysis. (**E**) Validation of SVEC4-10 XIAP^CRISPR cells with and without GFP-NEMO expression using western blot. Lysates were blotted for XIAP and Actin. (**F**) *Tnf, Kc,* and *Ccl5* expression of SVEC4-10 EMPTY^CRISPR and XIAP^CRISPR cells treated with 10 µM ABT-737, 10 µM S63845 and 30 µM Q-VD for 3 h. Graph is representative of three independent experiments. (**G**) Validation of NIK knockout in GFP-NEMO expressing SVEC4-10 EMPTY^CRISPR and NIK^CRISPR cells. Cells were treated for 2 h with 10 µM MG-132. Lysates were blotted for NIK and actin.

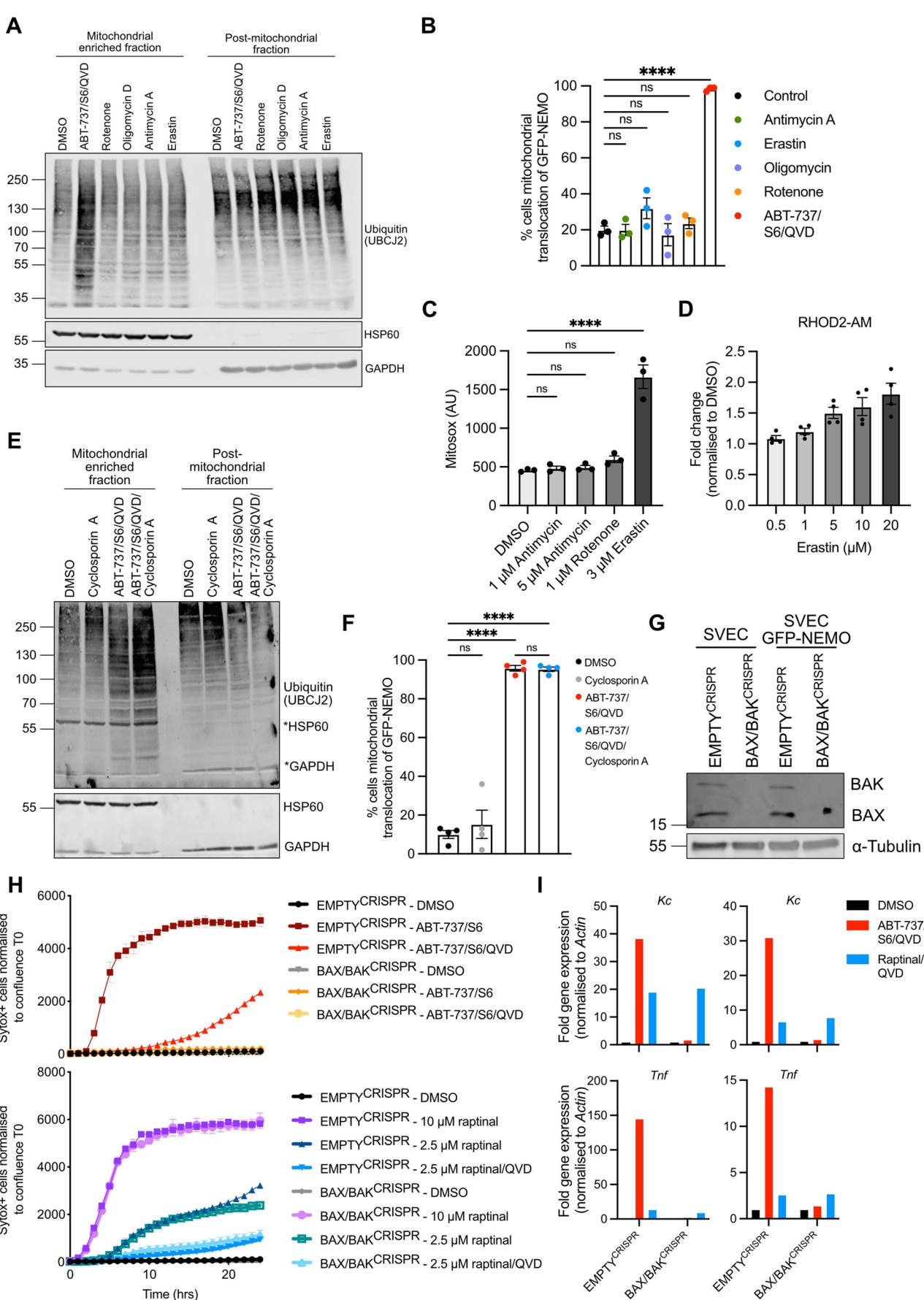

◄

**Figure EV5.  Raptinal induces cell death independent of mitochondrial permeabilization by BAX and BAK.**

(A) SVEC4-10 cells were treated for 2 h with 1 μM rotenone, 1 μM oligomycin, 5 μM antimycin A, 3 μM erastin or the combination 10 μM ABT-737, 10 μM S63845 and 30 μM Q-VD-OPh. Mitochondria were isolated using digitonin fractionation buffer and antibodies against ubiquitin (UBCJ2), HSP60 and GAPDH were used. (B) SVEC4-10 cells expressing GFP-NEMO were treated for 2 h with 1 μM rotenone, 1 μM oligomycin, 5 μM antimycin A, 3 μM erastin or the combination 10 μM ABT-737, 10 μM S63845 and 30 μM Q-VD-OPh. Cells were immunostained with mitochondrial TOM20 and DAPI for confocal microscopy. Graph shows the quantification of three independent experiments in which the percentage of cells with mitochondrial localisation of GFP-NEMO was analysed. (C) SVEC4-10 cells were treated for 2 h with 1 μM rotenone, 1 and 5 μM antimycin A or 3 μM erastin. ROS levels were determined using MitoSOX Red via flow cytometry. Graph displays mean values ± s.e.m. (error bars) of $n = 3$ independent experiments. (D) SVEC4-10 cells were treated with 0.5, 1, 5, 10 and 20 μM erastin. Mitochondrial calcium was measured using Rhod2-AM via flow cytometry. Graph displays mean values ± s.e.m. (error bars) of $n = 3$ independent experiments. (E) SVEC4-10 cells were treated for 2 h with 10 μM ABT-737, 10 μM S63845 and 30 μM Q-VD-OPh with or without 25 μM cyclosporin A. Mitochondria were isolated using digitonin fractionation buffer and antibodies against ubiquitin (UBCJ2), HSP60 and GADPH were used. (F) SVEC4-10 cells expressing GFP-NEMO were treated for 2 h with 10 μM ABT-737, 10 μM S63845 and 30 μM Q-VD-OPh with or without 25 μM cyclosporin A. Cells were immunostained with mitochondrial TOM20 and DAPI for confocal microscopy. Graph shows the quantification of three independent experiments showing the percentage of cells with mitochondrial localisation of GFP-NEMO, error bars represent s.e.m. (G) EMPTY[CRISPR] and BAX/BAK[CRISPR] validation of SVEC4-10 cells and SVEC4-10 cells expressing GFP-NEMO. Lysates for blotted for BAX, BAK and α-tubulin. (H) SVEC4-10 EMPTY[CRISPR] and SVEC4-10 BAX/BAK[CRISPR] cells treated with 10 μM ABT-737 and 10 μM S63845 or treated with 2.5 or 10 μM raptinal. Caspase-dependency of death was assessed using 30 μM Q-VD-OPh. Cell viability was measured using Sytox Green exclusion. Graphs are representative of two independent experiments and display the mean and s.e.m. of two replicates. (I) SVEC4-10 cells treated for 3 h with 10 μM ABT-737, 10 μM S63856 and 30 μM Q-VD-OPh or 2.5 μM raptinal and 30 μM Q-VD-OPh. Expression of *Kc*, *Tnf* and *Actin* were validated using RT-qPCR. Two repeats of Fig. 6E. Data information: (A, E) blots are representative for three independent experiments. Statistics performed using two-way ANOVA with Tukey correction. ***$P < 0.001$, ****$P < 0.0001$.

