## [Peer Review File · The EMBO Journal]

Mitochondrial outer membrane integrity regulates a ubiquitin-dependent and NF- κ B-mediated inflammatory response

Esmee Vringer, Rosalie Heilig, Joel Riley, Annabel Black, Catherine Cloix, George Skalka, Alfredo Montes-Gómez, Aurore Aguado, Sergio Lilla, Henning Walczak, Mads Gyrd-Hansen, Daniel Murphy, Danny Huang, Sara Zanivan, and Stephen Tait

Corresponding author: Stephen Tait (Stephen.Tait@glasgow.ac.uk)

Review Timeline:

Submission Date:	20th Feb 23
Editorial Decision:	9th Mar 23
Revision Received:	12th Nov 23
Editorial Decision:	13th Dec 23
Revision Received:	19th Dec 23
Editorial Decision:	5th Jan 24
Revision Received:	15th Jan 24
Accepted:	16th Jan 24

Editors: Karin Dumstrei / Ioannis Papaioannou

Transaction Report:

Dear Stephen,

Thank you for submitting your MS to The EMBO Journal. Your study has now been seen by two referees and their comments are provided below.

As you can see from the comments, the referees appreciate the added insight and is supportive of the work. They raise several constructive comments that should be sorted out in a revised version. Importantly we would also need some more mechanistic insight into the link between MOMP and NEMO translocation. We don't need the full mechanism, but some more insight is needed. Should you be able to address the raised concerns then I would like to invite you to submit a revised version.

It would be good to discuss the raised points further and I am available to do so via email or video.

I thank you for the opportunity to consider your work for publication. I look forward to discussing the revision plans further with you

best Karin

Karin Dumstrei, PhD
Senior Editor
The EMBO Journal

We realize that it is difficult to revise to a specific deadline. In the interest of protecting the conceptual advance provided by the work, we recommend a revision within 3 months (7th Jun 2023). Please discuss the revision progress ahead of this time with the

editor if you require more time to complete the revisions.

As a matter of policy, competing manuscripts published during this period will not negatively impact on our assessment of the conceptual advance presented by your study. However, we request that you contact the editor as soon as possible upon publication of any related work, to discuss how to proceed.

Referee #1:

The role of mitochondria in inflammation is now well established, and several mitochondria-generated pro-inflammatory stimuli have been described. This study identified two events that occur upon mitochondrial outer membrane permeabilization (MOMP), namely autophagy-independent loss of mitochondrial proteins (which is not investigated further in much detail) and the preferential ubiquitylation of numerous mitochondrial proteins in all mitochondrial compartments. The study further shows that this ubiquitylation triggers the mitochondrial recruitment of NEMO, which is linked to nuclear translocation of p65 and the induction of inflammatory genes.

I think that the basic finding of comprehensive ubiquitylation of mitochondrial proteins upon MOMP is very interesting, as it suggests that mitochondrial integrity is required to keep the organelle protected from what could be seen as a defense reaction. There are however a few issues regarding the downstream events that in my view should be worked out in more detail to justify the conclusions.

1. The loss of mitochondrial proteins by non-autophagic mechanisms is surprising and interesting but little is done in this study to understand or to document this effect further. In my view, this should be worked out some more, to understand it better and also to connect it to the rest of the story. I suggest two additional experiments: first, use the E1-inhibitor to test whether ubiquitination is required for protein loss. Secondly, I suggest at least a first assessment of the mitochondrial network by microscopy, as the authors have done throughout the manuscript, to see if there are obvious changes. The authors say mitochondria are degraded but this is not actually shown. In my view, these two experiments are necessary to enhance this part and link it to the rest of the study.
2. This study finds p65-activation upon MOMP through NEMO-recruitment to mitochondria. An effort has to be made to reconcile these new findings with the findings published earlier by the same laboratory. In their earlier paper (Giampazolias et al., NCB 2017), they showed that NF-kappa B-activation upon MOMP is the consequence of cIAP-antagonism and NIK-activation (i.e. alternative NF-kappaB although p65 was also activated). In the previous paper, it was found that e.g. TNF-induction was greatly reduced, and the nuclear translocation of p65 was reduced by about half in NIK-deficient cells. In the current study they focus on NEMO, which probably is not linked to NIK. It is certainly possible that several mechanisms operate but I think the previous publication cannot be ignored here. In their previous work, the authors have already generated NIK-deficient cells. Perhaps they could be used to test whether and how strongly NIK contributes to the downstream effects described here, for instance in the raptinal-setting.
3. Fig. 4G, H: it is convincing that mutant NEMO is not recruited to mitochondria but it is difficult to understand what is happening. As far as I understand it, the cells still express endogenous NEMO. Is it not recruited to mitochondria, and if not, why not? Do the mutants act as dominant negatives of NEMO-translocation? Why would that be the case? I think a satisfactory explanation should be sought.
4. Figure 4E is not very convincing. GFP-NEMO may be just highly enough expressed to permit assessment but the mutants do not seem to meet this criterion; the fluorescence is all over, including nucleus and the area outside the cells, presumably because the expression levels are so low. I can find no information how GFP-NEMO was expressed in the U2OS cells. Is this transient transfection, or are they stable cell lines? In my view, an effort has to be made to make this clearer. The shape of the mitochondrial network (judged by the TOM20 stain) also is quite different upon expression of NEMO-mutants. This is unexplained.
5. I think that the immunostaining should be controlled by a stain using an isotype control antibody. Perhaps that has been done but at least for the key stains this is an essential control.
6. Is the quantification of 'cells with mitochondrial ubiquitin' etc. done with ImageJ (i.e. objectively)? What was the threshold to count a cell as having this feature? It is not clear from the description in methods - please explain in more detail.

Minor

I think it is good practice to repeat experiments at least twice (Fig. 1C, 6B).

Please be clear what the symbols in the figures are. In Fig. 1E I take it that each symbol is one experiment but this should be stated.

Please detail what it means 'experiment performed with 4 independent repeats' (legend to Fig. 2). How were the results shown actually derived from the four repeats?

Fig. 6E, S4F: please show the individual results rather than a representative blot.

Further

It would be interesting to know whether any ubiquitin-modifying enzymes were identified at mitochondria. They may be ubiquitin-modified themselves, and perhaps we could learn more about the mechanism.

Referee #2:

Review Vringer et al., EMBOJ-2023-113825

Vringer et al. investigate the role of mitochondrial outer membrane permeabilization as an inducer of inflammation. Upon treatment with BH3-mimetics and inhibition of caspases, mitochondria undergo elimination in a Bax/Bak-dependent manner. This was associated with increased ubiquitination of the mitochondrial fraction but did not involve classic autophagy regulatory proteins. Ubiquitination was not restricted to the outer mitochondrial membrane but was also seen in the matrix. The observed type of ubiquitination was K63, while K48 showed a decrease. The recruitment of NEMO onto mitochondria was not dependent on PINK1. Neither could the authors detect a role for MAPL or MARCH5 in NF- κ B activation. The hypothesis that MOMP is the unique trigger for NEMO translocation is investigated. At this point, the data is clear but some further information on the role of MOMP is necessary to move this paper from a descriptive to a mechanistic study.

Specific Points:

1. The authors claim that Bax/Bak absence prevents mitochondrial ubiquitination in their setup. However, Figure S1C still shows ubiquitination signals in the Bax/Bak KO.
2. The extent of mitochondrial translocation of NEMO in Figure 4A is inconclusive and the authors cannot label it as "robust". In addition to showing IF, the authors should quantify NEMO translocation by biochemical methods, which are much more quantitative. This issue is even more pressing in Figure 4C, where the difference to cells not treated with the inhibitor is far less convincing than what the bar graph suggests.
3. What is the significance of MOMP in this mechanism? This mechanism is under the control of permeability pore oxidation, calcium flux and determines mitochondrial ATP output. In their manuscript, the authors provide little mechanistic insight into which of these is operating to trigger NEMO translocation and mitochondrial elimination. In this context, the drug raptinal remains largely a black box. In the original paper, it is claimed that raptinal releases cytochrome c, in the absence of a Bax/Bak pore. This unusual finding only adds to the mystery and here, the authors should look more into the reasons and not solely rely on this relatively poorly characterized compound.

Referee 1: *"I think that the basic finding of comprehensive ubiquitylation of mitochondrial proteins upon MOMP is very interesting, as it suggests that mitochondrial integrity is required to keep the organelle protected from what could be seen as a defense reaction. There are however a few issues regarding the downstream events that in my view should be worked out in more detail to justify the conclusions."*

Response: We thank the reviewer for their positive and constructive review, our responses to specific comments are below.

"1. The loss of mitochondrial proteins by non-autophagic mechanisms is surprising and interesting but little is done in this study to understand or to document this effect further. In my view, this should be worked out some more, to understand it better and also to connect it to the rest of the story. I suggest two additional experiments: first, use the E1-inhibitor to test whether ubiquitination is required for protein loss. Secondly, I suggest at least a first assessment of the mitochondrial network by microscopy, as the authors have done throughout the manuscript, to see if there are obvious changes. The authors say mitochondria are degraded but this is not actually shown. In my view, these two experiments are necessary to enhance this part and link it to the rest of the study. "

Response: We thank the reviewer for raising these important points, centring around the process(es) that mediate mitochondrial protein loss upon MOMP and assessment of mitochondrial morphology. As suggested, in new experiments, we investigated the impact of E1 inhibition (TAK243 treatment to inhibit ubiquitylation) upon mitochondrial protein loss, we additionally investigated the impact of proteasome inhibition, using MG132. As shown in Extended Figures 1F and H, loss of various mitochondrial proteins was observed at extended time points post-MOMP (24h, but not 8h (see Figure 1A)) and could be rescued by proteasome or E1 inhibition. Thus, ubiquitin-proteasome activity can contribute to loss of mitochondrial protein content at later timepoints following MOMP. Secondly, we investigated mitochondria morphology following MOMP, using MitoTracker Green FM, finding mitochondrial fragmentation and peri-nuclear aggregation at earlier time points (3h) following MOMP. At later time-points (24h) mitochondrial content was reduced (Figure 1C). We discuss these new data (lines 383 onwards), highlighting that recruitment of NEMO and inflammation engaged by MOMP occurs with significantly quicker kinetics relative to loss of mitochondria and mitochondrial protein degradation. This difference in kinetics suggests that mitochondrial degradation is unlikely to play an important, inhibitory role on the rapid inflammation engaged by MOMP.

"2. This study finds p65-activation upon MOMP through NEMO-recruitment to mitochondria. An effort has to be made to reconcile these new findings with the findings published earlier by the same laboratory. In their earlier paper (Giampazolias et al., NCB 2017), they showed that NF-kappa B-activation upon MOMP is the consequence of cIAP-antagonism and NIK-activation (i.e. alternative NF-kappaB although p65 was also activated). In the previous paper, it was found that e.g. TNF-induction was greatly reduced, and the nuclear translocation of p65 was reduced by about half in NIK-deficient cells. In the current study they focus on NEMO, which probably is not linked to NIK. It is certainly possible that several

mechanisms operate but I think the previous publication cannot be ignored here. In their previous work, the authors have already generated NIK-deficient cells. Perhaps they could be used to test whether and how strongly NIK contributes to the downstream effects described here, for instance in the raptinal-setting.”

Response: As the reviewer states, we previously identified a role for NIK in activating NF- κ B upon MOMP, the relationship of this (NIK activation) to NEMO-mitochondrial translocation was unclear. To investigate this, we used CRISPR-Cas9 to delete NIK in GFP-NEMO expressing SVEC cells, then determined GFP-NEMO translocation upon MOMP (Figures 5I and J, Supplementary Figure 4G). Importantly, we found no impact of NIK deletion upon NEMO mitochondrial translocation. This demonstrates that MOMP can elicit multiple pathways causing NF- κ B activation upon MOMP, consistent with this in our earlier publication residual NF- κ B activity even in the absence of NIK (Giampazolias *et al*, 2017). These new data are discussed lines 289 onwards.

“3. Fig. 4G, H: it is convincing that mutant NEMO is not recruited to mitochondria but it is difficult to understand what is happening. As far as I understand it, the cells still express endogenous NEMO. Is it not recruited to mitochondria, and if not, why not? Do the mutants act as dominant negatives of NEMO-translocation? Why would that be the case? I think a satisfactory explanation should be sought.”

Response: As highlighted by the reviewer, expression of mutant (non-ubiquitin binding) NEMO acts in a dominant negative manner to suppress NF- κ B activation in cells expressing endogenous wild-type NEMO. Others have previously shown that ubiquitin binding deficient NEMO retains the ability to key activating kinase, IKK beta (Ea *et al*, 2006; Wu *et al*, 2006), thus ectopic expression of mutant (non-ubiquitin binding, non-activatable) NEMO can sequester IKK beta preventing NF- κ B in cells expressing wild-type NEMO, underpinning this dominant negative effect.

“4. Figure 4E is not very convincing. GFP-NEMO may be just highly enough expressed to permit assessment but the mutants do not seem to meet this criterion; the fluorescence is all over, including nucleus and the area outside the cells, presumably because the expression levels are so low. I can find no information how GFP-NEMO was expressed in the U2OS cells. Is this transient transfection, or are they stable cell lines? In my view, an effort has to be made to make this clearer. The shape of the mitochondrial network (judged by the TOM20 stain) also is quite different upon expression of NEMO-mutants. This is unexplained.”

Response: Our apologies for the images of the ubiquitin binding GFP-NEMO mutants – all lines were generated through viral transduction, enabling stable expression. We have used alternative images (with increased expression) in the revised manuscript (Figure 4G, quantified in 4H). Consistent with our earlier data, we only observe mitochondrial localisation of GFP-NEMO upon a MOMP-inducing stimulus, demonstrating that NEMO’s ability to bind ubiquitin is key in its mitochondrial recruitment. The mitochondrial morphology is similar in all cases (fragmentation, perinuclear clustering) consistent with engagement of MOMP that we described earlier (see Figure 1C).

“5. I think that the immunostaining should be controlled by a stain using an isotype control antibody. Perhaps that has been done but at least for the key stains this is an essential control.”

Response: In new experiments. we used non-specific mouse and rabbit sera to serve as staining controls for anti-Ub (mouse) and TOM20 (rabbit antibodies). These data shown below (Reviewer Figure 1) show lack of staining (mouse serum) and low, diffuse staining (rabbit serum), supporting specific staining by the antibodies used in our study. Note that all antibodies used in this study were commercially sourced, precluding use of ideal control antibodies (e.g. pre-immune serum) for each individual antibody.

“6. Is the quantification of 'cells with mitochondrial ubiquitin' etc. done with ImageJ (i.e. objectively)? What was the threshold to count a cell as having this feature? It is not clear from the description in methods - please explain in more detail.”

Response: We have now extended our description of how cells were quantified in the materials and methods (line 588 onwards). Quantification of cells with mitochondrial ubiquitin or mitochondrial translocation of GFP-NEMO was obtained by manually counting the cells. Cells were counted using the Cell Count plugin in Fiji when there was overlap between the ubiquitin stain or GFP-NEMO with the mitochondrial stain. Different counters within the plugin were used to count all cells and cells showing overlap in mitochondrial and ubiquitin or GFP-NEMO. Images from 1 experiment were analysed in one sitting for consistency. The same method was used for the quantification of nuclear p65.

Minor

“I think it is good practice to repeat experiments at least twice (Fig. 1C, 6B).”

Response: We thank the reviewer for raising this, presuming they meant at least three times (as opposed to twice above). Accordingly, the data shown in Fig 1C has been repeated a third time (now Fig 1D) with similar data as has the ubiquitin localisation in 6B (graph modified to incorporate this).

“Please be clear what the symbols in the figures are. In Fig. 1E I take it that each symbol is one experiment but this should be stated.”

Response: Apologies for the oversight these have now been defined

“Please detail what it means 'experiment performed with 4 independent repeats' (legend to Fig. 2). How were the results shown actually derived from the four repeats?”

Response:

Data shown are averaged from 4 biological repeats. Mass spectrometry analysis is performed in the same run for all samples. Volcano plot of ubiquitylated proteins in SVEC4-10 cells treated for 3 hours with 10 μ M ABT-737, 10 μ M S63845 and 30 μ M Q-VD-OPh. Changes in ubiquitinated sites intensities were determined using a permutation-based

Student's t test with a 1% FDR. Only significantly changed ubiquitinated sites were coloured indicating gain and loss of intensity.

"Fig. 6E, S4F: please show the individual results rather than a representative blot."

Response: We now show all individual results (in addition to ones shown) in Extended Figure 5I, consistently showing induction of inflammatory cytokines by raptinal independent of BAX and BAK, in contrast to BAX/BAK dependent activation of inflammation by BH3-mimetics.

Further

"It would be interesting to know whether any ubiquitin-modifying enzymes were identified at mitochondria. They may be ubiquitin-modified themselves, and perhaps we could learn more about the mechanism."

Response: In agreement with the reviewer's suggestion, we had previously noted both MUL1 and MARCH5 (E3 mitochondrial resident ubiquitin ligases) were ubiquitylated upon MOMP (please see Supplemental Table 1), this suggested to us that they may be important for the promiscuous ubiquitylation that we detected – we now note this point at the relevant section (investigating possible role for MUL1/MARCH5 in mitochondrial ubiquitylation) (line 274). However our finding that deletion of both MUL1/MARCH5, does not impact mitochondrial ubiquitylation suggesting that both E3 ligases are targets of ubiquitylation as opposed to being actively involved in mitochondrial ubiquitylation.

Referee 2: *"Vringer et al. investigate the role of mitochondrial outer membrane permeabilization as an inducer of inflammation. Upon treatment with BH3-mimetics and inhibition of caspases, mitochondria undergo elimination in a Bax/Bak-dependent manner. This was associated with increased ubiquitination of the mitochondrial fraction but did not involve classic autophagy regulatory proteins. Ubiquitination was not restricted to the outer mitochondrial membrane but was also seen in the matrix. The observed type of ubiquitination was K63, while K48 showed a decrease. The recruitment of NEMO onto mitochondria was not dependent on PINK1. Neither could the authors detect a role for MAPL or MARCH5 in NF- κ B activation. The hypothesis that MOMP is the unique trigger for NEMO translocation is investigated. At this point, the data is clear but some further information on the role of MOMP is necessary to move this paper from a descriptive to a mechanistic study."*

Response: We very much appreciate the constructive critique, our responses to specific comments are below.

Specific Points:

1. The authors claim that Bax/Bak absence prevents mitochondrial ubiquitination in their setup. However, Figure S1C still shows ubiquitination signals in the Bax/Bak KO.

Response: We thank the reviewer for highlighting this, to clarify, BAX/BAK deletion prevents the MOMP induced increased in mitochondrial ubiquitylation, not basal mitochondrial ubiquitylation that is not MOMP-regulated, but presumably occurs through various E3 ligases, for instance MARCH5 and MAPL. We have modified our text accordingly.

2. The extent of mitochondrial translocation of NEMO in Figure 4A is inconclusive and the authors cannot label it as "robust". In addition to showing IF, the authors should quantify NEMO translocation by biochemical methods, which are much more quantitative. This issue is even more pressing in Figure 4C, where the difference to cells not treated with the inhibitor is far less convincing than what the bar graph suggests.

Response: As suggested the reviewer we have attempted to determine the extent of NEMO translocation through biochemical approaches including conventional dounce-isolation of mitochondrial fractions (Reviewer Figure 2) or through rapid immunopurification of mitochondria using HA-tagged OMP25 (Reviewer Figure 3). In the case of conventional mitochondrial isolation we were unable to detect an increase in mitochondrial NEMO (over background), in immunopurified mitochondria we were unable to detect NEMO, these data suggest that mitochondrial association of NEMO is only evident in intact cells. Potentially the difficulty in detecting stable NEMO/mitochondrial interactions relates to the reduced affinity of NEMO for K63-linked ubiquitin (Lo *et al*, 2009; Rahighi *et al*, 2009). We have removed the word "robust" since it is subjective. In new experiments, we investigated the kinetics of NEMO mitochondrial translocation through live-cell imaging, this shows rapid mitochondrial translocation occurring within 60 minutes of BH3-mimetic addition (Figures 4A and B, Movie 2).

"3. What is the significance of MOMP in this mechanism? This mechanism is under the control of permeability pore oxidation, calcium flux and determines mitochondrial ATP output. In their manuscript, the authors provide little mechanistic insight into which of these is operating to trigger NEMO translocation and mitochondrial elimination. In this context, the drug raptinal remains largely a black box. In the original paper, it is claimed that raptinal releases cytochrome c, in the absence of a Bax/Bak pore. This unusual finding only adds to the mystery and here, the authors should look more into the reasons and not solely rely on this relatively poorly characterized compound."

Response: The reviewer raises the important question as to how MOMP causes mitochondrial ubiquitylation and NEMO recruitment. To investigate this, in new experiments, we determined whether cellular effects associated with MOMP – mitochondrial permeability transition pore, reactive oxygen species generation, loss of mitochondrial respiratory function and mitochondrial calcium uptake – in themselves lead to mitochondrial ubiquitylation and NEMO recruitment (Extended Figures 5A - E). None of these processes were found to cause mitochondrial ubiquitylation, NEMO recruitment. These new data are discussed line 301 onwards. While they don't identify how MOMP leads to mitochondrial ubiquitylation, NEMO recruitment, the data strongly argue against roles for established MOMP initiated effects. We discuss this new data line 371 onwards, speculating that exposure of inner membrane specific protein(s) may be a driving event in this process, something we are further investigating.

Reviewer Figure 1 Confocal analysis of control or CICD treated cells with ubiquitin, TOM20 and control non-specific mouse and rabbit sera

SVEC4-10 cells were treated with DMSO or 10 μ M ABT-737, 10 μ M S63845 and 30 μ M Q-VD-OPh for 3 hours. Cells were fixed and stained for ubiquitin (FK2), mitochondria (TOM20, rabbit) and mouse or rabbit IgG isotype control. Images were taken using the Nikon A1R, 63x oil objective. Scalebar is 20 μ m.

Reviewer Figure 2 Analysis of NEMO expression on mitochondria during CICD following dounce homogenisation

SVEC4-10 cells were treated with DMSO or 10 μ M ABT-737, 10 μ M S63845 and 30 μ M Q-VD-OPh for 2 hours. Following dounce homogenisation, inputs, mitochondria enriched and post-mitochondrial fraction were blotted for NEMO, HSP60 (mitochondrial matrix), actin and SMAC (intermembrane space protein degraded in cytosol upon MOMP).

Reviewer Figure 3: Analysis of NEMO expression on immunopurified mitochondria during CICD

SVEC4-10 cells expressing 3xHA-eGFP-OMP25 mitotag were treated with DMSO or 10 μ M ABT-737, 10 μ M S63845 and 30 μ M Q-VD-OPh for 2 hours. Following immunoprecipitation with HA beads, inputs, flow through (FT) and immunoprecipitates were blotted for NEMO and TOM20. Methodology adapted from (Chen *et al*, 2016).

References

- Chen WW, Freinkman E, Wang T, Birsoy K, Sabatini DM (2016) Absolute Quantification of Matrix Metabolites Reveals the Dynamics of Mitochondrial Metabolism. *Cell* 166: 1324-1337 e1311
- Ea CK, Deng L, Xia ZP, Pineda G, Chen ZJ (2006) Activation of IKK by TNFalpha requires site-specific ubiquitination of RIP1 and polyubiquitin binding by NEMO. *Mol Cell* 22: 245-257
- Giampazolias E, Zunino B, Dhayade S, Bock F, Cloix C, Cao K, Roca A, Lopez J, Ichim G, Proics E *et al* (2017) Mitochondrial permeabilization engages NF-kappaB-dependent anti-tumour activity under caspase deficiency. *Nat Cell Biol* 19: 1116-1129
- Lo YC, Lin SC, Rospigliosi CC, Conze DB, Wu CJ, Ashwell JD, Eliezer D, Wu H (2009) Structural basis for recognition of diubiquitins by NEMO. *Mol Cell* 33: 602-615
- Rahighi S, Ikeda F, Kawasaki M, Akutsu M, Suzuki N, Kato R, Kensche T, Uejima T, Bloor S, Komander D *et al* (2009) Specific recognition of linear ubiquitin chains by NEMO is important for NF-kappaB activation. *Cell* 136: 1098-1109
- Wu CJ, Conze DB, Li T, Srinivasula SM, Ashwell JD (2006) Sensing of Lys 63-linked polyubiquitination by NEMO is a key event in NF-kappaB activation [corrected]. *Nat Cell Biol* 8: 398-406

Dear Stephen,

Thank you for the submission of your revised manuscript to The EMBO Journal. We have now received the comments of the two referees that were asked to re-evaluate your study (included below). As you will see, both are satisfied with the revision and your responses to their previous concerns, and they now support publication of the study. There is only one remaining suggestion (by ref. #2) regarding the addition of a figure showing the quantification results of the NEMO signal increase on homogenized mitochondria, which would further strengthen the manuscript.

From the editorial side, there are also a few minor changes that we need from you before we can proceed with acceptance of the manuscript:

- Please provide up to 5 keywords in your revised manuscript (after the Abstract).
- Please make sure that the deposited datasets will be publicly available at the time of publication and add the specific URLs to the Data availability statement. The reviewer access information can now be removed.
- Please enter all relevant funding information in our online manuscript handling system. It should match exactly the information provided in the Acknowledgements section of your manuscript.
- Please note that we request authors to consider both actual and perceived competing interests (you can review our policy here: <https://www.embopress.org/page/journal/14602075/authorguide#conflictsofinterest>) and declare them in a "Disclosure and competing interests statement" at the end of Materials and Methods.
- Please add the heading "References" before your list of citations.
- The movie legends should be removed from the manuscript file; instead, please provide each legend in a Word/text file zipped together with the corresponding movie file. The nomenclature also needs correction to "Movie EV1" and "Movie EV2" throughout the manuscript.
- The supplementary pdf file with Supplemental Table 1 should be named "Appendix"; this should have a brief Table of Contents including page numbers on its first page. Please submit this file in pdf format. The table in this file should be renamed "Appendix Table S1" (please also update its callouts in the manuscript file accordingly). Please move the legend of this table from the manuscript file to the Appendix.
- Please upload your synopsis image in jpg or png format. The final dimensions of this image must be 550 pixels (width) x 300-600 pixels (height), and all text should be legible at the final size.
- Figure callouts for Fig. 2C,D are missing; please note that all figures and their panels should be called out in your revised manuscript (in alphabetical order).
- Please consider adding a separate "Data Information" section at the end of each Figure legend for reporting information related to data, sample sizes, statistics, representation (e.g. error bars, scale bars) etc. Please see our guide for more information on Figure legends: <https://www.embopress.org/page/journal/14602075/authorguide#figureformat>
- Please indicate the statistical test used for data analysis in the legends of Figures 2a, d; EV3b.
- Please note that information related to the sample size/number of replicates (n) is missing in the legends of Figures 1f; 2a; 3c, e; 4d, f, h, j; 5c, f, h, j; EV2d; EV3b, e.
- Please note that the error bars are not defined in the legends of Figures 1b, f; 3c, e; 4b, d, f, h, j; 5c, f, h, j; 6b, d; EV1a; EV2d; EV3b, e; EV5c, d, f.

As soon as these issues are resolved, I might contact you again to discuss with you a few suggestions for minor improvements in the title, abstract and synopsis text.

Please also note that as part of the EMBO publications' Transparent Editorial Process, The EMBO Journal publishes online a Peer Review File along with each accepted manuscript. This File will be published in conjunction with your paper and will include the referee reports, your point-by-point response and all pertinent correspondence relating to the manuscript. You can opt out of this by letting the editorial office know (contact@embojournal.org). If you do opt out, the Peer Review File link will point to the following statement: "No Peer Review File is available with this article, as the authors have chosen not to make the review

process public in this case."

We look forward to seeing a final version of your manuscript as soon as possible. Please use this link to submit your revision:
<https://emboj.msubmit.net/cgi-bin/main.plex>

Best regards,

Ioannis

Referee #1:

The role of mitochondria as triggers of inflammation is increasingly appreciated, and a number of signals have been identified that drive this. The activation of NF- κ B has been the focus on I think four reports now, all describing different mechanisms. While revision of this manuscript was underway, one paper was published (PMID: 37683611), which also described recruitment of NEMO to damaged mitochondria. However, that paper investigated damage not through apoptosis (but through chemical depolarisation) and found an essential role of Parkin, which this study explicitly excludes. This study goes way beyond anything published previously.

I think this is an important study. It highlights an important and novel pro-inflammatory mechanism of mitochondria. Even though this mechanism likely operates only when caspases are not activated, it is very relevant to know. I have raised my concerns in my first review. The authors have not addressed every point but I am entirely satisfied by their response and the additional experimentation as shown.

Referee #2:

The authors have mostly addressed my concerns and the ones of the other reviewer. However, in my opinion, the dounce-homogenized mitochondria show increased NEMO signals on mitochondria, which I estimate about double of the baseline. The authors should generate a figure with $n=3$ to quantify this increase, as it provides an alternative assay, thus making their statements more convincing.

The authors addressed the editorial issues.

Dear Stephen,

Thank you for the submission of your revised manuscript to The EMBO Journal and for satisfactorily addressing most of our previous requests. There are a few minor issues that remain to be resolved before we can proceed with publication of the manuscript:

- Please address the remaining point of referee #2 (regarding quantification of the NEMO signal increase in mitochondria following treatment) by adding appropriate data/figures in your revised manuscript and commenting on the results in a response to the referee. If you add new main or EV Figure panels, please update the Figure callouts (throughout the manuscript) as needed and also upload the respective source data for the new panels.
- Please indicate the statistical test used for data analysis in the legend of Figure 2d.
- Please note that the error bars are not clearly defined in the legends of Figures 6b, d.
- We recommend adding a separate "Data Information" section at the end of each Figure legend for avoiding redundancy in reporting information related to data, sample sizes, statistical tests, representation (e.g. error bars, scale bars) etc. Please see our guide for more information and examples of Figure legends:
<https://www.embopress.org/page/journal/14602075/authorguide#figureformat>
- Please move the legend of Appendix Table S1 (formerly "Supplemental Table 1") from the main manuscript file to the Appendix.

We look forward to seeing a final version of your manuscript as soon as possible. Please use this link to submit your revision:
<https://emboj.msubmit.net/cgi-bin/main.plex>.

Best regards,

Ioannis

The authors addressed the remaining editorial issues.

Reviewer 2: The authors have mostly addressed my concerns and the ones of the other reviewer. However, in my opinion, the dounce-homogenized mitochondria show increased NEMO signals on mitochondria, which I estimate about double of the baseline. The authors should generate a figure with n=3 to quantify this increase, as it provides an alternative assay, thus making their statements more convincing.

Response: We have repeated the expt. two further times and include the individual blots in the revised version (Figure 4C and EV2A). As noted by the reviewer in all repeats we detect an increase in mitochondrial-localised NEMO, in line with our imaging data.

Dear Stephen,

I am pleased to inform you that your manuscript has been accepted for publication in The EMBO Journal.

Best regards,

Ioannis
